# Nunataks as barriers to ice flow: implications for palaeo ice-sheet reconstructions

Martim Mas e Braga[1,2], Richard Selwyn Jones[3,4], Jennifer C. H. Newall[1,2], Irina Rogozhina[5], Jane L. Andersen[6], Nathaniel A. Lifton[7,8], and Arjen P. Stroeven[1,2]

[1]Geomorphology & Glaciology, Department of Physical Geography, Stockholm University, Stockholm, Sweden
[2]Bolin Centre for Climate Research, Stockholm University, Stockholm, Sweden
[3]Department of Geography, Durham University, Durham, UK
[4]School of Earth, Atmosphere and Environment, Monash University, Melbourne, Australia
[5]Department of Geography, Norwegian University of Science and Technology, Trondheim, Norway
[6]Department of Geoscience, Aarhus University, Aarhus, Denmark
[7]Department of Earth, Atmospheric, and Planetary Sciences, Purdue University, West Lafayette, USA
[8]Department of Physics and Astronomy, Purdue University, West Lafayette, USA

**Correspondence:** Martim Mas e Braga (martim.braga@natgeo.su.se)

**Abstract.** Numerical models predict that discharge from the polar ice sheets will become the largest contributor to sea-level rise over the coming centuries. However, the predicted amount of ice discharge and associated thinning depends on how well ice sheet models reproduce glaciological processes, such as ice flow in regions of large topographic relief, where ice flows around bedrock summits (i.e. nunataks) and through outlet glaciers. The ability of ice sheet models to capture long-term ice loss is best

tested by comparing model simulations against geological data. A benchmark for such models is ice surface elevation change, which has been constrained empirically at nunataks and along margins of outlet glaciers using cosmogenic exposure dating. However, the usefulness of this approach in quantifying ice sheet thinning relies on how well such records represent changes in regional ice surface elevation. Here we examine how ice surface elevations respond to the presence of strong topographic relief that acts as an obstacle by modeling ice flow around and between idealised nunataks during periods of imposed ice sheet

thinning. We find that, for realistic Antarctic conditions, a single nunatak can exert an impact on ice thickness over 20 km away from its summit, with its most prominent effect being a local increase (decrease) of the ice surface elevation of hundreds of metres upstream (downstream) of the obstacle. A direct consequence of this differential surface response for cosmogenic exposure dating is a delay in the time of bedrock exposure upstream relative to downstream of a nunatak summit. A nunatak elongated transverse to ice flow is able to increase ice retention and therefore impose steeper ice surface gradients, while

efficient ice drainage through outlet glaciers produces gentler gradients. Such differences, however, are not typically captured by continent-wide ice sheet models due to their coarse grid resolutions. Their inability to capture site-specific surface elevation changes appears to be a key reason for the observed mismatches between the timing of ice free conditions from cosmogenic exposure dating and model simulations. We conclude that a model grid refinement over complex topography and information about sample position relative to ice flow near the nunatak are necessary to improve data-model comparisons of ice surface elevation, and therefore the ability of models to simulate ice discharge in regions of large topographic relief.

# 1 Introduction

Ongoing changes in climate are already causing significant mass loss and ice-margin retreat of both the Antarctic and Greenland ice sheets (Garbe et al., 2020; King et al., 2020). Near-future (2100 CE) projections of sea-level rise point to ocean thermal expansion as the main cause (Oppenheimer et al., 2019), but over multi-centennial timescales, the sea level contribution from Antarctica is expected to become dominant (Pattyn and Morlighem, 2020). Numerical ice sheet modelling efforts are aimed at reducing uncertainty by better understanding the processes that lead to sea-level rise, focusing on both shorter (Goelzer et al., 2020; Seroussi et al., 2020), and longer timescales (Pollard and DeConto, 2009; Albrecht et al., 2020). Recent efforts include improvements in key model components such as grounding line dynamics (e.g. Gladstone et al., 2017; Seroussi and Morlighem, 2018), coupling to solid Earth and sea level models (e.g. Gomez et al., 2020), and improved treatment of ice-ocean interaction processes (e.g. Reese et al., 2018; Kreuzer et al., 2020). The importance of bedrock topography (Morlighem et al., 2020) and grid resolution (Durand et al., 2011) have been acknowledged previously, and studied particularly for marginal regions of the ice sheet (e.g. Sun et al., 2014; Robel et al., 2016; Favier et al., 2016). Spatial variations in bedrock topography, and the resulting basal and lateral drag exerted at the ice-bedrock interface for different spatial scales, can slow down or even stabilise grounding line retreat (Jamieson et al., 2012, 2014; Åkesson et al., 2018; Jones et al., 2021; Robel et al., In Review). Regions near the ice sheet margin with large subglacial topographic relief, such as the overridden mountain ranges that fringe the glaciated cratons of Greenland and East Antarctica (Howat et al., 2014; Burton-Johnson et al., 2016), therefore require suitable consideration when evaluating ice loss beyond this century.

The accuracy of ice sheet models is limited by grid resolution (e.g. Cuzzone et al., 2019), simplifications in model physics (Hindmarsh, 2004; Cuffey and Paterson, 2010), and uncertainties in the climate forcing (e.g. Seguinot et al., 2014; Alder and Hostetler, 2019; Niu et al., 2019; Mas e Braga et al., 2021). To improve their predictive ability, models require validation with empirical observations. When considering changes on multi-centennial timescales, ice sheet reconstructions and ice thinning trends in the geological past (over hundreds to thousands of years) provide useful bounds on potential ice sheet change, and important observations for constraining and testing model simulations. By providing empirical targets for ice-sheet model simulations, uncertain model parameters can then be fine-tuned, and a closer match between data and models can be achieved (e.g. Golledge et al., 2012; Seguinot et al., 2016; Patton et al., 2017). However, interpreting such data in terms of properly providing constraints to ice sheet models, and using ice sheet models to better interpret the data, requires a better understanding of the interaction between ice flow and complex subglacial terrain.

Exposed bedrock, especially where mountain summits extend through the ice (i.e., nunataks), provide suitable targets for cosmogenic nuclide surface exposure dating. Determining the time of bedrock exposure allows inferring when the ice surface last reached the sampled elevation. In Antarctica, most exposed bedrock is situated in regions of large topographic relief (Fig. 1), and this is the predominant setting from which rock samples are acquired (e.g. Ackert et al., 1999; Lilly et al., 2010; Bentley et al., 2014; Suganuma et al., 2014; Jones et al., 2017). Cosmogenic nuclide exposure ages are determined by the concentrations of cosmogenic nuclides in erratic boulders or cobbles (i.e. glacially transported clasts of a different lithology than the underlying bedrock, deposited during periods of ice cover) or bedrock surfaces. The concentration of a cosmogenic

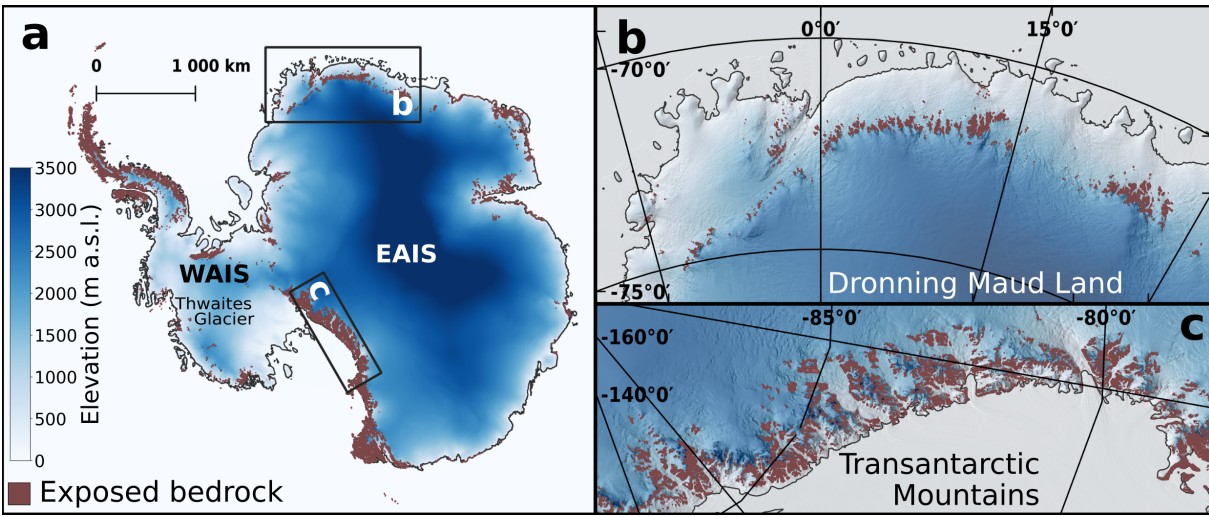

**Figure 1.** Present-day ice surface elevation (in m above sea level, m a.s.l.) in Antarctica (Morlighem et al., 2020). (a) the grounded part of the Antarctic Ice Sheet, including two example settings where the East Antarctic Ice Sheet (EAIS) overrides steep subglacial terrain of marginal mountain ranges in Dronning Maud Land and the Transantarctic Mountains (panels b and c, respectively), and Thwaites Glacier for reference; (b) Dronning Maud Land, where individual nunataks and nunatak ranges tower above the ice surface; and (c) the Transantarctic Mountains, where outlet glaciers are laterally confined by nunatak ranges. Brown areas denote exposed bedrock. Shading on panels (b) and (c) highlights the steep surface topography.

nuclide increases the longer a rock surface is exposed to cosmic rays (Gosse and Phillips, 2001). Assuming no cosmogenic nuclides have accumulated during a previous period of exposure (i.e., an assumption of no inheritance), and using known nuclide-specific production rates, a measured cosmogenic nuclide concentration can be interpreted in terms of the last time the ice sheet covered that specific location, and consequently the ice surface elevation at the time of sample exposure or deposition. The elevation of a sample above the present-day ice surface then enables determination of the magnitude of ice thinning between the time of exposure and the present. Yet, the gradient in ice surface elevation up- and downstream of a nunatak range often does not have a clear established relationship with the regional ice surface, a deficiency highlighted by Andersen et al. (2020). This is because *(i)* exposure ages only provide local constraints regarding ice surface elevation at a particular time, and *(ii)* considering the elevation gradient between upstream and downstream ice surfaces, and an expectation that these gradients will change as the ice sheet thins, it is unclear how a rock sample elevation can be consistently related to a representative regional ice surface elevation. Furthermore, when compiling cosmogenic-nuclide exposure ages sampled around the entire Antarctic continent, it appears that there is no systematic approach to selecting the sampling position on a nunatak relative to ice flow direction around the nunatak (Fig. 2), highlighting that so far this problem has received little attention.

While ice sheet models of Greenland and Antarctica have been able to broadly fit ice geometries reconstructed from empirical data, including the approximate rates of ice thinning that are recorded by cosmogenic exposure ages (e.g., Whitehouse et al., 2012; Briggs et al., 2014; Albrecht et al., 2020), most models struggle to replicate the inferred timing of ice thickness change

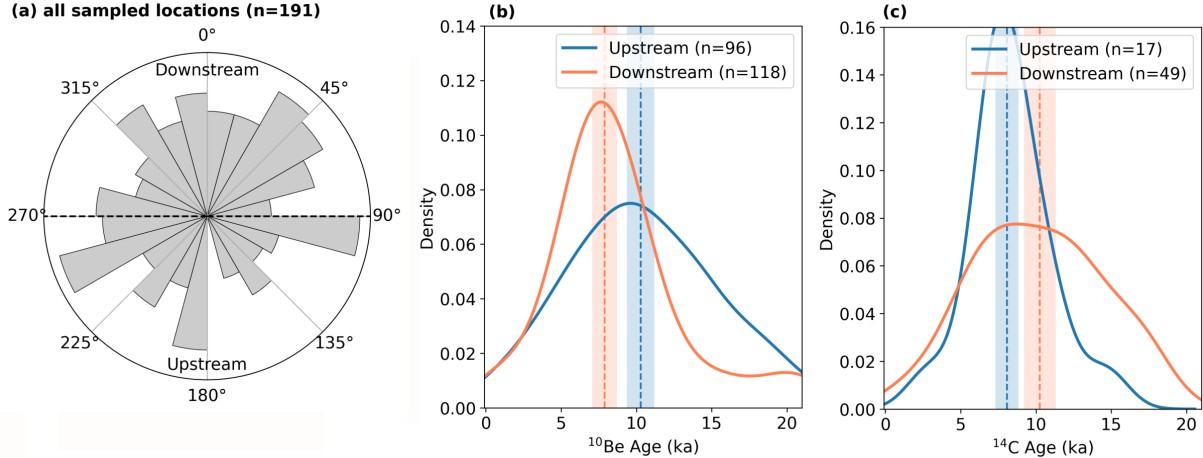

**Figure 2.** (a) Polar histogram showing the location of cosmogenic $^{10}$Be and $^{14}$C samples from boulders in Antarctica (Heyman, 2021; Balco, 2021) with ages younger than the Last Glacial Maximum, relative to the nearest nunatak summit and its adjacent ice flow direction (n = 191; sample duplicates were excluded). The difference in direction was computed between a sample position relative to the nunatak summit (identified in BedMachine-Antarctica; Morlighem et al., 2020) and ice flow (Mouginot et al., 2012) near the nunatak summit. Summits were identified through a morphological feature map (Wood, 1996, see supplementary material). The area of each slice is proportional to the number of samples within that category, and each category spans a 15°arc. In this figure, 0°(180°) implies that the sample was taken downstream (upstream), directly aligned with the ice flow direction. (b,c) kernel density function of the $^{10}$Be (b) and $^{14}$C ages (c) apparent exposure ages from samples shown in panel (a). Dashed lines show the median age, and shading shows the uncertainty interval based on the median uncertainty of their respective ages.

(Jones et al., 2020; Stutz et al., 2020; Johnson et al., 2021). Such data-model mismatches are likely due to a combination of factors, one of which is the spatial resolution of the models (Lowry et al., 2020; Johnson et al., 2021). When run over a glacial-interglacial cycle, ice sheet models do not typically resolve the pattern of ice flow around individual nunataks, and consequently cannot resolve the transient response of the ice surface at the sampled locations. We address this limitation through two tests.

First, we test the supposition that an ice sheet up and downstream of a nunatak experiences different degrees of thinning due to the interaction of ice flow with an obstacle. Second, we explore whether a model with horizontal grid resolutions comparable to those currently employed by large-scale ice sheet models (5–20 km) capture this interaction. To perform these tests, we apply a numerical ice flow model to an idealised bedrock topography typical of Antarctic settings.

## 2   Data and Methods

To better understand how ice flow interacts with the steep topography of nunataks and what impact this interaction has on ice surface elevation patterns, we perform a suite of numerical simulations using an idealised setup. We use Úa (Gudmundsson, 2020), an ice flow model that solves the shallow shelf, also referred to as shelfy stream approximation (SSA or SStA) of the

Stokes equations (Cuffey and Paterson, 2010) on a horizontal, finite element mesh. Úa has been successfully applied to model the ice flow of idealised (e.g. Gudmundsson et al., 2012), modern (e.g. Miles et al., 2021), and palaeo-ice streams (e.g. Jones et al., 2021). Úa solves the ice surface and momentum equations simultaneously, and its finite element formulation allows for an adaptive mesh refinement in areas of particular interest, such as where the ice shallows around nunataks. For the modelled domain (which we describe below), an unstructured mesh was generated, which is refined during simulation time (including during spin up) based on a series of glaciological refinement criteria. Element size is refined according to ice thickness, from a maximum of 8 km down to 205 m around the interface between ice and nunatak, where ice thickness approaches the minimum (which we set as 1 m). The mesh is also refined (to 500 m) in a 4 km buffer zone centered at the grounding line. The mean (median) element size after spin up was 740 m (414 m).

In the vicinity of nunataks, where the ice is thinnest, the ice flow regime has a relevant vertical component, which is not captured by the SSA approximation. A more accurate representation of the ice dynamics for such regions can be achieved with full-Stokes models. However, such models are currently too computationally demanding when adopted for long simulation periods and multiple experiments (e.g. Schannwell et al., 2020). Over the thinnest-ice areas, the horizontal scales resolved in our model are of the order of a few hundred metres (commonly noted as $O(10^2)$ m), while the vertical scale is of the order of a few metres ($O(10^0)$ m). This yields an aspect ratio of $O(10^{-2})$, which falls within the range where shallow approximations are applicable ($O(\leq 10^{-2})$, e.g. Fowler and Larson, 1978). To our knowledge, no intercomparisons between simplified and full-Stokes models exist that focus on the representation of thickness gradients. At least within the context of MISMIP+ (Cornford et al., 2020), there are little variations between SSA (including the model used in our study), L1Lx, and higher-order models regarding simulated ice retreat. Full-Stokes models that participated in the MISMIP+ experiment also agreed with the simplified-physics models, indicating that other models should behave similarly. Instead, the MISMIP+ experiments highlighted the importance of the formulation of the sliding law (i.e., Weertman versus Coulomb-limited). The two sliding laws strongly differed in their rates of grounding line retreat, but it has been shown that such differences decrease with increasing spatial resolution (i.e., model grid/mesh refinement) at the grounding line (Gladstone et al., 2017). Here, we adopt a Weertman law for basal sliding and the required refinement at the grounding line to minimise sliding-law issues.

At the free upper surface, the streamline upwind Petrov-Galerkin (SUPG) method is applied, which ensures model stability over regions of pronounced ice surface topography (Wirbel and Jarosch, 2020). In our experiments, the ice surface topography is steepest when the ice front retreats from the downstream end of the domain and in the vicinity of nunataks. When ice thins below the prescribed minimum (which we set to 1 m), the model uses the active-set method (Durand et al., 2009; Gudmundsson et al., 2012). In this method, violated ice thickness constraints are activated using the Lagrange multiplier approach (Ito and Kunisch, 2008), which is applied to the momentum equation, and ensures a better representation of the ice dynamics compared to simply resetting the thickness to the prescribed minimum value, as is commonly used in finite element models.

The model domain (Sect. 2.1) and spin up procedure (Sect. 2.2) are the same for all simulations. In a first set of simulations, we evaluate changes in ice surface elevation up and downstream of a single nunatak under three different thinning scenarios (Sect. 2.3). We then use the forcing that provides the highest ice-thinning rates to evaluate the impact of multiple nunataks, and the width of glaciers between them, on ice surface elevation patterns (Sect. 2.4). Finally, we repeat the last experiments

for a series of regular meshes (without refinement) at horizontal resolutions commonly used in ice sheet models (Sect. 2.5). This final set of experiments assesses how well different grid resolutions resolve changes in ice surface elevation across steep
ice-marginal topography under thinning scenarios, and their implications for simulations of past ice surface elevations.

## 2.1 Model domain setup

The model domain consists of a $300 \times 200$ km rectangular section of an idealised ice sheet. The domain coordinates extend from $x = -150$ km to $x = 150$ km, and $y = -100$ km to $y = 100$ km. We create an ice cap mirrored along the x direction, i.e. where the centre of the domain ($x = 0$) represents the ice divide. We apply all changes symmetrically with respect to $x$, but
focus our analysis on the positive side of the domain (i.e. $x > 0$, thus our effective domain is 150 km long). This is done in order to ensure a natural boundary at the upstream side (the ice divide). The downstream side is kept as an open boundary, allowing the ice front to retreat. On the lateral limits, a free-slip boundary condition is applied (i.e. velocities are zero perpendicular to the boundary, but unconstrained parallel to it), and unless stated otherwise, we only consider the region within 50 km of the centreline ($y = 0$) to avoid boundary effects.
We set the bed elevation at the divide ($B_0$) to 750 m above sea level (m a.s.l.), which we keep constant in all experiments. We prescribe a prograde sloping bed ($B$) with an inclination $\beta = 0.9$ % (Eq. 1), which results in the bed sloping below sea level at $x \approx 83$ km.

$$B(x) = B_0 - \beta \cdot |x| \tag{1}$$

Because nunataks close to the Antarctic coast, or along the margins of palaeo ice streams, are often situated in proximity to
over-deepened troughs, which can trigger Marine Ice Sheet Instability, a retrograde-slope case was also tested. In this case, we set $B_0$ to $-750$ m and $\beta$ to $-0.1$ %, so that the bedrock at the downstream end of the domain is at the same elevation (-600 m a.s.l.) in both types of experiments, which is representative of continental shelf depths of Antarctica (Morlighem et al., 2020). The grounding line in Úa is defined by the flotation condition, and although its migration is not the focus of our study, we allow it to move freely, refining the mesh elements around the grounding line as it evolves (as described at the end of this section).
Nunataks in this domain are represented as three-dimensional Gaussian surfaces, which are superimposed on B with their centre (i.e. the nunatak summit) at $|x| = 50$ km and $y = 0$ km. All generated nunataks have an outcrop size (i.e. exposed area above the ice surface) after spin up of approximately $12 \times 5$ km along its main axes. Depending on the experiment, the nunatak was elongated transverse to flow (i.e. along the y axis) or parallel to flow (i.e. along the x axis; Fig. 3). The adopted idealised nunatak dimensions and the value of $\beta$ are within the interval constrained by 33 real nunatak topographic profiles
across Antarctica (Figs. S1–S3).

The effects of ice rheology (including temperature) in Úa are accounted for by the strain-rate factor $A$ in Glen's flow law. To compute $A$, which we treat as spatially uniform and constant over time for the purpose of our experiments, we assume an ice temperature (T) of $-20$ °C. This yields a value similar to that found for regions surrounding nunatak escarpments in Antarctica, as obtained in Gudmundsson et al. (2019) when inverting for $A$ and basal slipperiness ($C$) based on satellite-derived ice surface

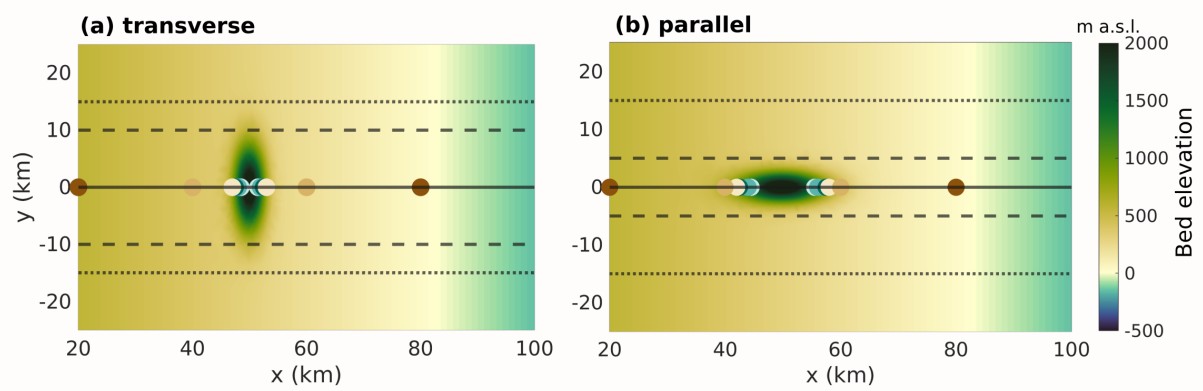

**Figure 3.** Bedrock elevation (m a.s.l.) used in the experiments where the nunatak is placed (a) transverse and (b) parallel to ice flow. Ice flow in this figure is from left to right. Transect lines show the position of the transects presented in Fig. 5a,b, and coloured circles show the locations for the ice surface evolution analysis presented in Fig. 5c,d following the same colours as the lines therein.

velocities. Following the same reasoning for $A$, we assume basal sliding to follow Weertman's sliding law (Weertman, 1957), and use a constant value for $C$ of $log_{10}(C) \approx -4.5$ ($C = 2.9 \cdot 10^{-5}$ m kPa$^{-1}$a$^{-1}$). The main model parameters are summarised in Table 1, and sensitivity analyses of values for A and C are presented in the supplementary material (Figs. S4, S5).

The surface mass balance (SMB; in metres per year, ma$^{-1}$) parameterisation applied to the spin up and subsequent experiments is given by $a(x,t)$ in Eq. (2):

$$a(x,t) = a_0 - \frac{a_0 - a_e}{L_x} \cdot |x| + b(t) \tag{2}$$

where $a_0$ is the SMB at the ice divide ($x = 0$), $a_e$ is the SMB at the edge of the domain ($x = 150$ km), $L_x = 150$ km, and $b(t)$ is a time-dependent SMB term, through which perturbations to the total SMB are applied. We prescribe $a_0 = 1.3$ ma$^{-1}$, and $a_e = -0.3$ ma$^{-1}$, which results in no ablation occurring over our region of interest (black line in Fig. 4a), as usual for Antarctic settings (Agosta et al., 2019).

**2.2   Model spin up**

The model spin up starts from a 1200 m-thick uniform ice distribution, to which a constant (but spatially variable) SMB is applied (i.e. $b(t) = 0$ ma$^{-1}$ in Eq. 2) for 20 thousand years (kyr). This period is long enough for the system to reach equilibrium with the SMB forcing. The average ice surface slope between the ice divide and the grounding line (which after spin up was located at $x = 136$ km; Fig. 4b), is $\sim 1.3$ %. This inclination is representative of measured profiles along nunataks for various regions of the Antarctic Ice Sheet (Figs. 4d, S1, S2). The resulting surface velocity varies from 0 ma$^{-1}$ at the divide to 121

ma$^{-1}$ downstream of the grounding line, with median and mean velocities of 29 and 33 ma$^{-1}$, respectively (see Fig. S9 for a comparison of velocity profiles along the centreline). The ice flow velocity increases along the nunatak flanks, with a maximum

**Table 1.** Model parameters used in this study. Values for $B_0$ and $\beta$ reflect values for prograde and retrograde slopes, respectively.

| Parameter | Value | Units |
|---|---|---|
| Basal Slipperiness (C) | $2.9 \cdot 10^{-5}$ | $\mathrm{m\,kPa^{-1}a^{-1}}$ |
| Weertman's sliding law $m$ exponent | 3 | |
| Ice temperature [for $A = A(T)$] | $-20$ | $^{o}$C |
| Glen's flow law $n$ exponent | 3 | |
| Ice density ($\rho_i$) | 910 | $\mathrm{kg\,m^{-3}}$ |
| Sea water density ($\rho_w$) | 1028 | $\mathrm{kg\,m^{-3}}$ |
| Gravity (g) | 9.81 | $\mathrm{m\,s^{-2}}$ |
| Ice-divide SMB ($a_0$) | 1.3 | $\mathrm{m\,a^{-1}}$ |
| Domain-end SMB ($a_e$) | $-0.3$ | $\mathrm{m\,a^{-1}}$ |
| Ice-divide bedrock elevation ($B_0$) | 750 and $-750$ | m a.s.l. |
| Bedrock inclination ($\beta$) | 0.9 and $-0.1$ | % |

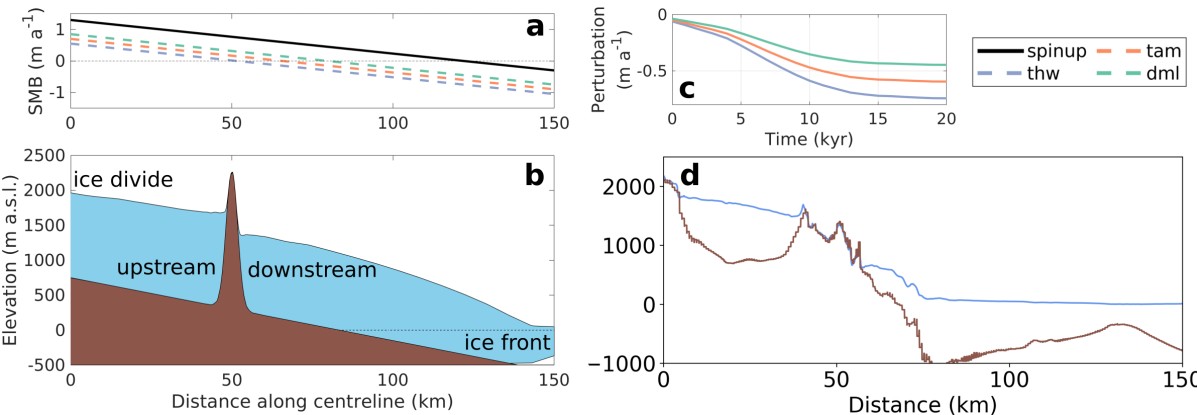

**Figure 4.** Model domain setup: (a) SMB ($\mathrm{m\,a^{-1}}$) profiles for the spin up and initial state (black), and at the end of each different ice thinning experiment ('thw', Thwaites; 'tam', Transantarctic Mountains; 'dml', Dronning Maud Land); (b) ice surface elevation (m a.s.l.) after the 20 kyr spin up for the across-flow elongated nunatak; (c) applied transient perturbation (in $\mathrm{m\,a^{-1}}$) to each ice thinning experiment during the 20 kyr simulations subsequent to spin up; (d) Bedrock and ice sheet surface elevation profiles across an example nunatak in Antarctica (Half-way nunatak, [78.38 °S, 161.1 °E], upper Skelton Glacier, Transantarctic Mountains; see Fig. S1 for another 32 examples).

of $\sim 53$ $\mathrm{ma^{-1}}$, consistent with observed values in Mouginot et al. (2012). Although velocities at the floating end are slower than observed for ice shelves in Antarctica, they are far from our region of interest, and the setup reproduces key features observed along the analysed nunatak profiles, such as ice surface gradients across nunataks and ice acceleration upstream and downstream of the nunatak (e.g. Figs. 4, S9). Our idealised setup focuses on capturing the key components mentioned above,

while excluding unnecessary complex features that could prevent identifying the ice surface response signal to the ice flow interaction with the obstacles.

## 2.3 Ice thinning experiments

In order to understand whether ice thinning occurs uniformly up and downstream of a nunatak, we impose three different degrees of ice thinning uniformly over the domain. The perturbations to SMB that induce ice thinning are applied through $b(t)$ in Eq. (2), and only evolve in one direction over the entire domain, towards increased ablation. The evolution of $b(t)$ is based on a smoothed step curve that applies a weight, evolving from 8 % to 100 %, mimicking the deglaciation progression recorded in ocean sediment and ice cores for the past 20 kyr (e.g. Lisiecki and Raymo, 2005; Jouzel et al., 2007). This "weight curve" is then multiplied by a constant chosen in order to match Last Glacial Maximum-to-present ice thinning at three contrasting regions as inferred from a continent-wide transient modelling experiment (Golledge et al., 2014). These regions are Thwaites Glacier, in West Antarctica ('thw', $-0.75$ ma$^{-1}$), and the Transantarctic Mountains ('tam', $-0.60$ ma$^{-1}$) and Dronning Maud Land ('dml', $-0.45$ ma$^{-1}$) in East Antarctica (Fig. 4c). Using the 'thw' scenario as an example, the value for $b(0) = -0.75 \cdot 0.08$ ma$^{-1}$, and $b(20kyr) = -0.75 \cdot 1.0$ ma$^{-1}$. These three different thinning scenarios were applied to nunataks that were elongated along and transverse to ice flow, and to reference experiments without a nunatak for comparison purposes.

## 2.4 Three-nunatak experiments

Across much of Antarctica and Greenland, nunataks are more common in groups than in isolated cases. The aim of this set of experiments is therefore to test whether three nunataks, separated by narrow glaciers, yield ice surface elevations and gradients that differ from the control run due to a combined effect of all nunataks on ice flow. For this test, we perform sets of experiments with the same SMB as applied in the 'thw' experiment with one nunatak (control run), but using three nunataks aligned transverse to flow at $|x| = 50$ km and of the same size as in the ice thinning experiments. The spacing between nunataks differs in each experiment, resulting in glacier widths of 0 (i.e. the three nunataks form a single, wider barrier), 5, 10 and 15 km. The range of applied widths reflects realistic values observed around Antarctica (Fig. S3c; Howat et al., 2019).

## 2.5 Mesh-resolution experiments

In a final series of sensitivity experiments, we assess how well regular-spaced grids of coarser resolutions typically used in continental-scale ice sheet models (5, 10, and 20 km) resolve the ice surface elevation pattern around nunataks compared to the solution using an unstructured, locally refined mesh. We do so by repeating the full set of three-nunatak experiments and the one-nunatak 'thw' scenario (as control) for these regular meshes (i.e. without refinement).

All sets of experiments, their respective surface mass balance (SMB) perturbations, and the number of nunataks in each set are summarised in Table 2.

**Table 2.** Summary of the experiments performed in this study. Adaptive-mesh experiments are those that have a mesh refinement of up to 205 m around nunataks and the grounding line. Regular-mesh experiments have no refinement anywhere in the domain.

| Experiment set | # Nunataks | # Experiments | Nunatak aspect | b(t) at t=20 kyr [ma$^{-1}$] | Mesh type |
|---|---|---|---|---|---|
| spinup | 0, 1 | 3 | elongated across and along flow, and no nunatak | 0.0 | refined |
| thw | 0, 1 | 3 | same as above | -0.75 | refined |
| tam | 0, 1 | 3 | same as above | -0.60 | refined |
| dml | 0, 1 | 3 | same as above | -0.45 | refined |
| thw0y1n_mshXXkm | 1 | 3, with XX = [5, 10, 20] km mesh resolution | elongated across flow | -0.75 | regular XX km |
| thw0y3nXXkm | 3 | 4, with XX = [0, 5, 10, 15] km glacier widths | same as above | -0.75 | refined |
| thw0y3nXXkm_msh05km | 3 | 4, same as above | same as above | -0.75 | regular 5 km |
| thw0y3nXXkm_msh10km | 3 | 4, same as above | same as above | -0.75 | regular 10 km |
| thw0y3nXXkm_msh20km | 3 | 4, same as above | same as above | -0.75 | regular 20 km |

## 3   Results

### 3.1   Ice thinning experiments

Our experiments clearly demonstrate that the presence of a nunatak impacts ice surface elevations, and that the response magnitude depends on its orientation relative to ice flow. The flow barrier created by the nunatak causes ice to accumulate upstream, depleting the downstream side of ice and causing the ice surface there to lower (Figs. 5a,b). At the start of the thinning experiments where the nunatak is placed transverse to flow, the effect of the nunatak on the ice surface elevation is seen up to 30 km away from its summit along flow, and 15 km perpendicular to flow. The ice surface directly upstream of the nunatak is 360 m above the general ice sheet surface (considered to be the elevation 15 km away from the centreline), while downstream the lowest ice surface elevation (located 3 km downstream of the summit) is 100 m lower than the general ice surface elevation. Along the nunatak flanks, the ice surface elevation falls about 300 m. In summary, while the general ice surface slope over the grounded part of the domain is 1.3 %, the slope is about 2–3 times steeper around the nunataks. For experiments where the nunatak is elongated parallel to ice flow, the impact of the nunatak is smaller, producing a lower ice surface elevation gradient, and relative elevation-change effects that can be discerned 20 km up and downstream along the centreline, and up to 5 km transverse to the centreline (Fig. 5b). In both cases, changes in ice surface elevation perpendicular to ice flow, caused by the nunatak presence, stand out from the general ice surface elevation for the entire extent of the subglacial nunatak obstruction (cf. Figs. 5a,b, and 3a,b).

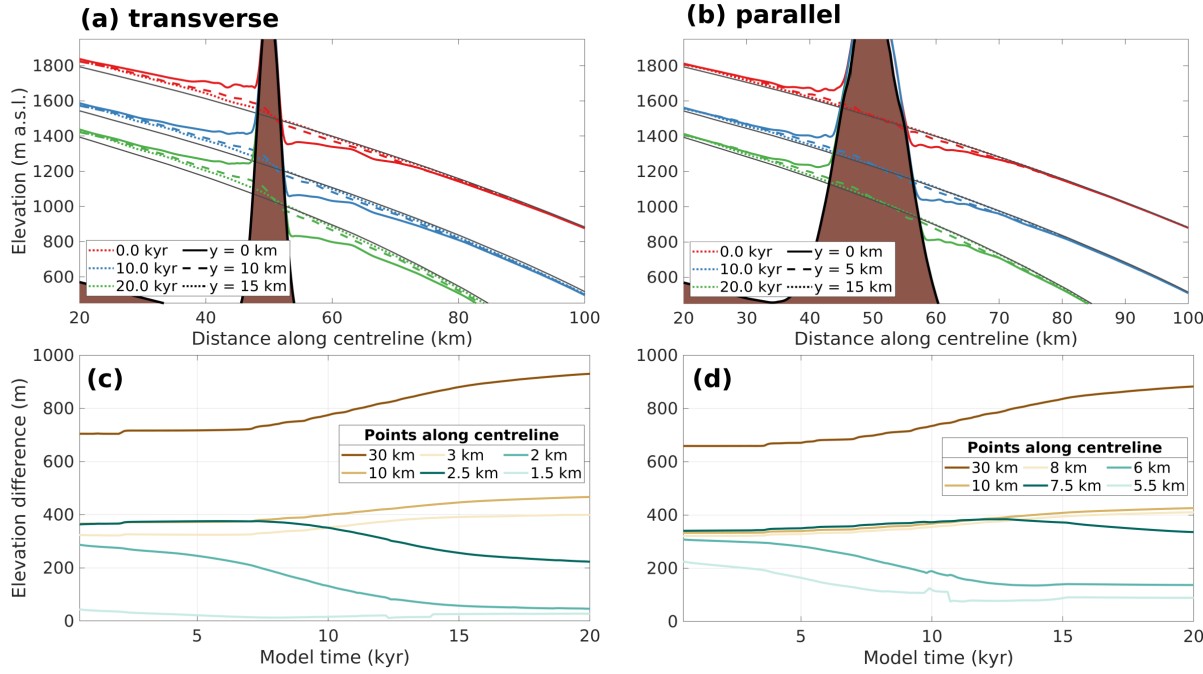

**Figure 5.** Surface elevations after 0, 10, and 20 kyr for the 'thw' experiment with a nunatak elongated (a) transverse and (b) parallel to ice flow at 0, 10, and 15 km from the centreline; dark grey lines show the respective ice surface elevation for experiments without a nunatak; (c,d) evolution of the ice surface elevation difference between six pairs of equidistant points up and downstream from the nunatak, along the centre line for the experiments showcased in (a,b) respectively. In short, this figure shows to which extent a single nunatak is able to influence ice surface elevation in space (along and perpendicular to flow) and through time, how ice surface elevation is linked to a nunatak's subglacial topography, and how the ice surface elevation mismatch evolves differently depending on the distance from the nunatak.

Both ice thinning experiments detailed in Fig. 5a,b reveal an overall steepening of the ice surface gradient during the 20 kyr
thinning period (profiles from red to green). The evolution of this steepening is illustrated in Fig. 5c,d, where points equidistant from the nunatak summit (upstream and downstream) show ice surface elevation differences that increase through time. For equidistant locations closer to the nunatak summit (< 2.5 km and < 7.5 km for Fig. 5c and d, respectively) the increase in ice surface steepening (i.e., in elevation difference) is disrupted when the downstream location becomes exposed, since its elevation no longer changes while the equivalent point upstream is still ice covered and thinning. This pattern of increased
steepening around nunataks is potentially important for cosmogenic nuclide studies to consider because from these modeling experiments, significant differences in the time of exposure of up and downstream faces of nunataks become apparent (Fig. 6).

To analyse the difference in timing of ice surface evolution and nunatak surface exposure up and downstream of the nunatak, we select five pairs of points equidistant from the nunatak summit. They span a distance from where the downstream side is already exposed at the start of the experiment (1.5 km) to the closest element to the nunatak where the ice continues to thin
normally (i.e. it does not become exposed or stops thinning; 2.5 km) in any ice-thinning scenario. The chosen points (Fig. 6)

are also separated in the model mesh by one element size from one another. In our analysis, we consider that nunatak surface exposure commences when ice thickness falls below 10 m. A thickness threshold larger than the minimum thickness in the model is used for a consistent identification of the timing of surface exposure between the different experiments and different points analysed, and 10 m yielded the best results among the thresholds tested. For the purpose of cosmogenic exposure dating, most cosmogenic nuclide production occurs when ice is thinner than ~10 m (Gosse and Phillips, 2001). At the upstream side, if the bedrock surface remains ice covered according to the previous criterion, we then determine when the ice surface stabilises and thus reaches its minimum elevation. The implications of comparing time of exposure (thickness < 10 m) with time of stabilisation (thinning < 5 cm/100 yrs) are discussed further in Sect. 4.1. For visualisation purposes, only the experiments with nunataks elongated transverse to ice flow are shown in Fig. 6, but the patterns shown also hold true for nunataks elongated parallel to ice flow. Because of the elongation of the nunatak, its initial area exposed along the centreline is larger, and thus the pair of points compared for the latter are located further from the nunatak summit.

The differences in the ice surface elevation between the up and downstream sides of the nunatak result in differences in the time of nunatak surface exposure. This occurs for all scenarios, but with different lags in the time of exposure depending on the degree of thinning and distance from the summit. For some scenarios and locations, only the downstream side is exposed, while its upstream counterpart is still thinning at the end of the simulation time. Within 2 km from the nunatak summit (or 6 km for the nunatak elongated parallel to flow), and for those scenarios where both up and downstream sides are exposed (or meet the stabilisation criterion), the upstream side lags its downstream counterpart by 2 to 14 kyr (Fig. 6b). In a retrograde-slope setting, however, this effect is not observed. Rapid and uniform thinning happens once accelerated grounding line retreat is triggered, akin to Marine Ice Sheet Instability, yielding a similar adjustment up and downstream of the nunatak (Fig. S10).

## 3.2 Three-nunatak experiments

The experiments with three nunataks show how different glacier widths produce dissimilar responses of ice surface elevation (Fig. 7), mimicking situations where multiple nunataks are separated by narrow glaciers. After 20 kyr of ice thinning, the 0 km-width experiment (i.e. where the nunataks merge into a single, 36 km-wide nunatak; Fig. 7a) yields the highest ice retention upstream (further delaying surface lowering at the upstream side), with its surface up to 250 m higher compared to the control one-nunatak case (Fig. 7e). For the experiments with the smallest glacier widths (0, 5 km; Figs. 7a, b), the constructive interference between the nunataks results in an ice retention that is strong enough to instigate a strong deficit response downstream. Compared to the control scenario, this response reaches as far as 50 km downstream of the nunataks, also impacting the grounding line position, displacing it further inland on the lee side of the nunatak range. For the experiments where glacier widths are 10 and 15 km (Figs. 7c, d), the influence of multiple nunataks decreases but similar patterns arise, albeit with smaller differences compared with the control experiment.

The differences in glacier width also result in different organisations of ice flow around the nunataks. The wider barriers formed by the 0 and 5 km-wide glaciers yield a different pattern of ice flux upstream and downstream of the nunataks (cf. Figs. 7a,b and 8), where most of the flux is concentrated 13–30 km away from the centreline. In the case of the two experiments that have wider glaciers (10 and 15 km), ice flux also peaks further away from the centreline than in the control experiment,

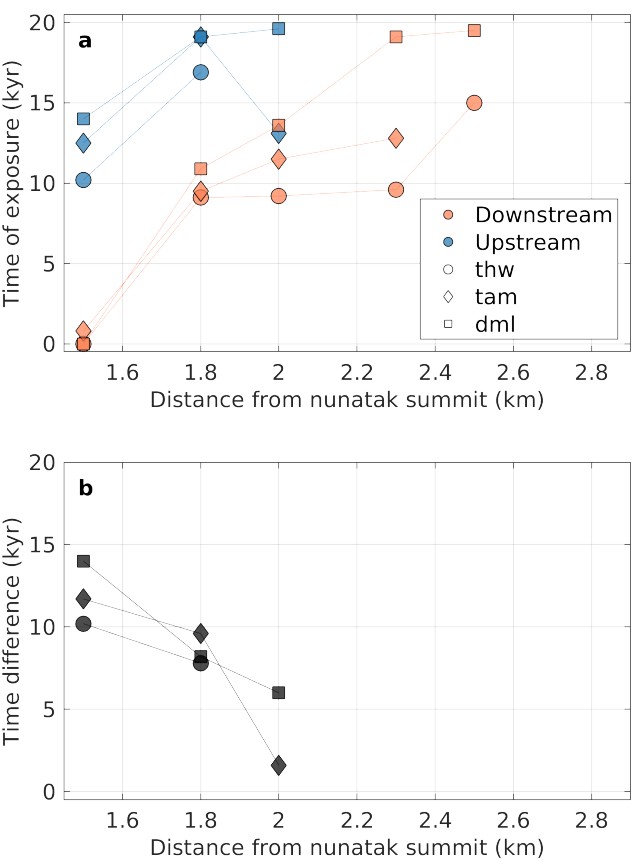

**Figure 6.** Relationship between distance from the nunatak summit and time of exposure (or stabilisation of the ice surface; model kyr) for nunataks elongated transverse to ice flow under the different thinning scenarios (see Table 2). (a) Time of exposure (in model years) for upstream (blue) and downstream (orange) points for the different thinning experiments: 'thw' (circles), 'tam' (diamonds), and 'dml' (squares). (b) As in panel (a), but showing the time difference between equivalent points for the cases where exposure or stabilisation happens for both up and downstream points. This figure shows how upstream points lag their respective downstream counterparts (by up to 14 kyr) in all thinning scenarios.

but is more uniformly distributed across the domain. Their largest flux peaks are closer to those of the control experiment, and distinct smaller peaks occur exactly where the glaciers are located. The difference between the flux downstream and upstream is inversely proportional to glacier width (i.e. the narrower the glacier, the larger the difference), which points to an increased retention of ice upstream as the cause for the increase in relative heightening/lowering of the ice surface.

### 3.3 Mesh-resolution experiments

The coarser regular-mesh experiments (5, 10, and 20 km horizontal resolution) applied to the series of three-nunatak configurations show important differences from the refined-mesh experiments (Fig. 9, cf. Fig. 7). The 5 km-mesh experiments deviate the

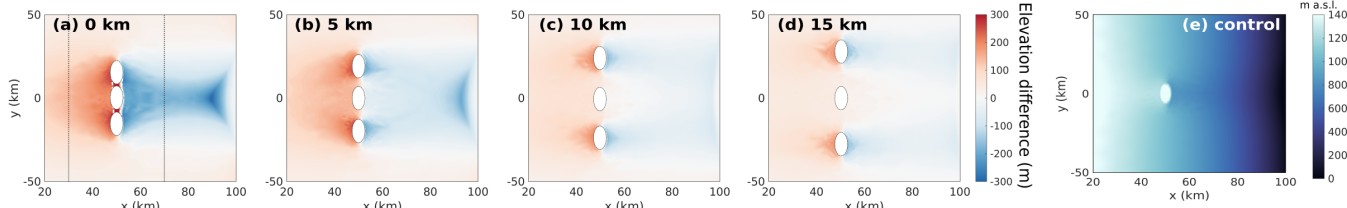

**Figure 7.** (a–d) Difference in surface elevation (m) between each three-nunatak 'thw' experiment of varying glacier widths (0, 5, 10, 15 km, respectively) with respect to the one-nunatak 'thw' control run (shown in panel e for reference), after 20 kyr of simulation. White elipses delineate the nunataks in each experiment, and dashed white lines in panel (a) enclose the region between them where bedrock is exposed during this experiment. Dotted black lines in panel (a) illustrate the position of the transects shown in Fig. 8. From this figure it is clear that a larger obstacle or a narrow glacier can increase the differential response up and downstream of the nunataks when compared to the single-nunatak case.

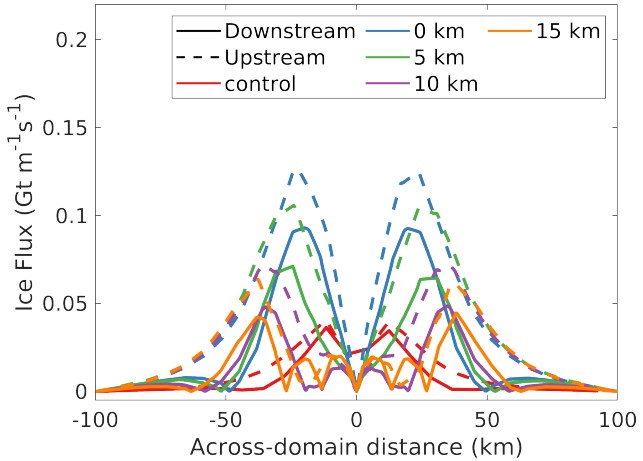

**Figure 8.** Ice flux (in $\mathrm{Gt\,m^{-1}s^{-1}}$) across transects 20 km upstream (dashed lines) and 20 km downstream (solid lines) of the nunatak summits ($x = 50$ km), as illustrated in Fig. 7a. Line colors denote the varying widths of glaciers separating the three nunataks, while 'control' (red line) refers to the one-nunatak experiment. Note how the experiment where glaciers are 15 km-wide yields a profile closest to the control experiment.

least from the refined-mesh experiments, and capture the relationship between glacier width and ice surface elevation gradient. Still, this mesh does not capture the same magnitude of ice surface lowering downstream, underestimating it by as much as 80 % (100–200 m) relative to the refined mesh, an effect that is particularly pronounced close to the nunataks (cf. Figs. 9a,b and Figs. 7a,b).

The 10 km mesh captures, to some degree, the original nunatak shape and consequent surface elevation increases upstream and decreases downstream of the nunataks (Figs. 9f–j). However, there are problems with both signal strength (under/overestimations of at most 250 m) and location (not coinciding with the pattern seen in the refined-mesh experiments). In

the 20 km mesh, the original nunatak shape is not properly captured (Fig. 9k–o), and the large element sizes cause increased heightening/lowering of ice surface elevation to occur much farther away up and downstream of the obstacles compared to their respective refined-mesh counterparts (Fig. 7). Finally, the 20 km mesh yields similar results for the 0, 5, and 10 km glacier-width experiments (Figs. 9k–m), but yields a much different response for the 15 km-width experiment (Fig. 9n), which is much more in tune with the results of the refined mesh. The differences between the regular-mesh and the refined-mesh experiments also result in different thinning rates between them (Fig. 10). The 20 km resolution model run overestimates the rate of thinning under the same forcing (Fig. 10c), while the 5 km resolution model run shows much closer values to the refined-mesh experiment, despite the underestimated thinning (Fig. 10b).

The finest regular mesh tested (5 km) performs the best among the regular meshes, since it partially captures the ice flow through the widest glaciers tested (15 km wide), and best represents the effect of glacier width on ice flow constriction (Fig. 11). The differences between ice flow downstream and upstream shown in Fig. 11 evolve similarly in the refined and 5 km meshes, but are not well represented in the 10 or 20 km meshes. The coarsest mesh (20 km) shows similar results for the 5 and 15 km-wide glacier experiments, and for the wider nunatak (0 km) and the 10 km-wide glacier experiments. This is because the coarse mesh resolution misses two nunataks summits, which lie between two nodes. As a result, the lower topography that is captured by this coarse mesh becomes a subglacial continuation of the single central nunatak. This also explains why the 15 km-width experiment shows an ice surface elevation pattern similar to the control run. Further tests indicate that the 20 km mesh only captures the existence of three nunatak summits when they are spaced by glaciers that are at least 20 km wide (not shown). Still, in these tests the glaciers are not wide enough for the mesh to capture their existence, and thus the results only reflect the effect of a wider obstacle.

## 4 Discussion

### 4.1 Ice surface response to ice flow around nunataks and through narrow glaciers

Our modelling experiments demonstrate that the magnitude of the ice surface elevation response due to the presence of nunataks is proportional to their ability to obstruct or constrict ice flow. The nunatak orientation relative to ice flow, or in the experiments where 3 nunataks are present, the width of the glaciers formed between them, modulate the advection of ice downstream, and consequently the ice surface elevation difference. Jamieson et al. (2014) showed that much wider channels ($\sim 40$ km), more characteristic of ice streams, can already provide lateral drag capable of decreasing advection of ice downstream, slowing down unstable grounding line retreat. In their fjord experiment, Frank et al. (2021) used widths that are more similar to our narrower glaciers ($\sim 5$ km), and showed that advection of ice from wider to narrower passages, as happens in our experiments, can be greatly slowed down by the lateral drag provided by the nunatak flanks. The magnitude of this response in our experiments could have been influenced by our use of uniform and constant ice rheology, which results in more rigid ice in the narrow glaciers, where flow is faster (e.g. Minchew et al., 2018). The experiments indicate that, across ice flow (i.e. along the $y$ direction), the extent of the ice surface that is impacted is more likely related to the interaction of ice flow with the subglacial extension of the nunatak, as commonly reported for different spatial and temporal scales, and modelling setups of different

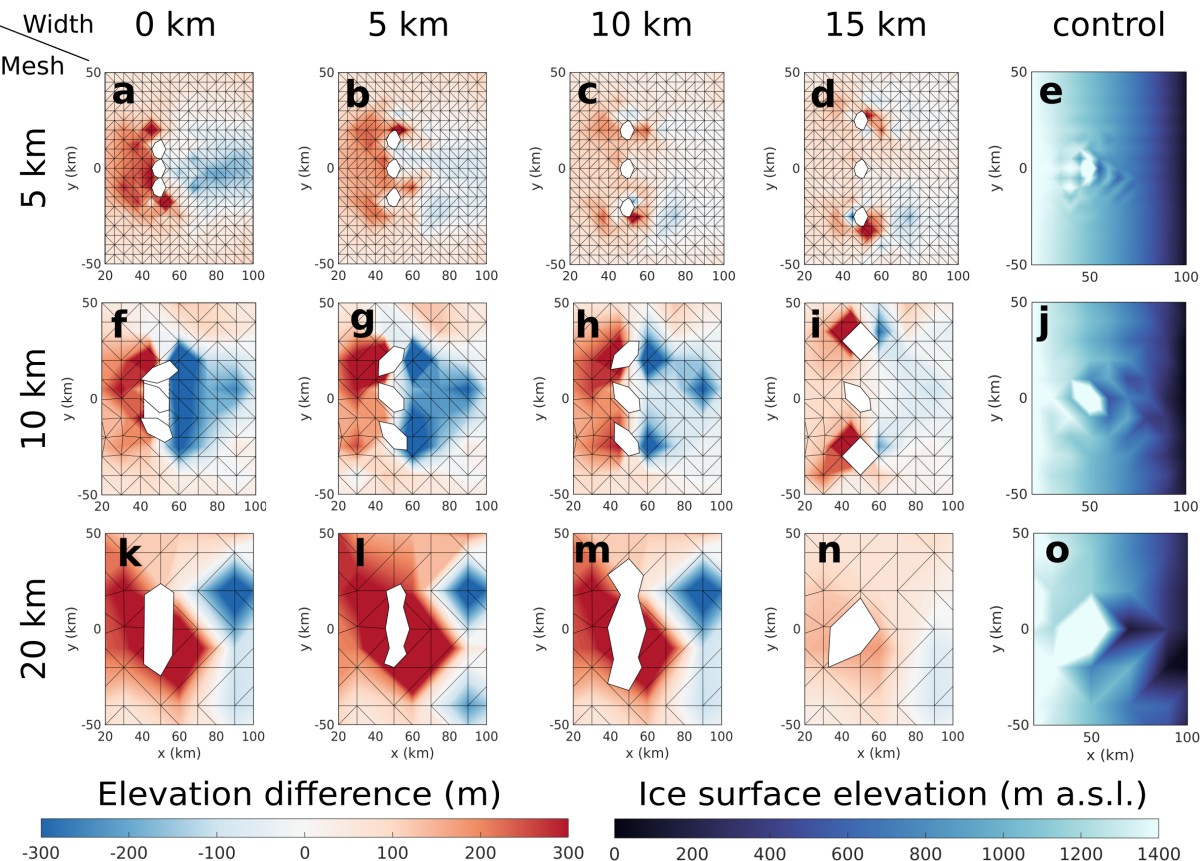

**Figure 9.** Difference in surface elevation (m, as in Fig. 7) between each 'thw' three-nunatak configuration (varying glacier widths) and the 'thw' control with one nunatak (right-most panels), for three regular mesh resolutions of 5, 10, and 20 km. The reference experiments were also performed on a regular mesh. These figures illustrate how differential elevation changes up and downstream of the nunataks (digitised in white based on their outcrop size, linearly interpolating the ice surface between the nodes and vertices) differ from the experiments using a refined mesh (Fig. 7), and how this mismatch decreases with increasing glacier width.

complexities (e.g. Siegert et al., 2005; Durand et al., 2011; Cuzzone et al., 2019; Paxman et al., 2020). A relative increase in the ice surface elevation was observed immediately downstream of the nunatak, while the lowest elevation attained was located 3 km downstream of the nunatak summit at the start of the simulations. Although a similar effect can be observed on the lee

side of isolated nunataks in Dronning Maud Land, MacRobertson Land, and the Transantarctic Mountains (Howat et al., 2019; Andersen et al., 2020), we cannot rule out that this is a numerical effect, caused by artificial advection of ice downstream from the regions where the minimum thickness constraint is violated and thus numerically modified (e.g., Jarosch et al., 2013).

The ice surface steepening and consequent mismatch between the up and downstream sides of the nunatak increases as the ice thins, until the downstream side becomes exposed. Exposure happens earlier downstream, as expected due to lower ice

surface elevation, and an equidistant point upstream becomes exposed (or has its thinning stabilised) up to 14 kyr later than

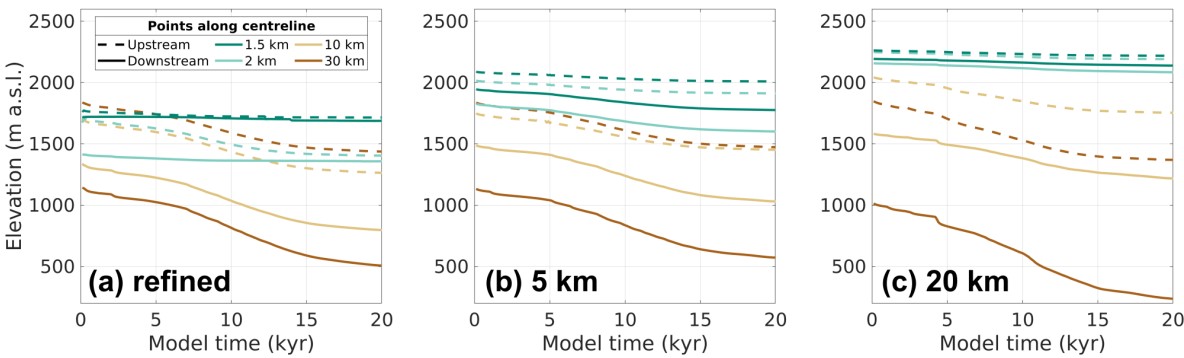

**Figure 10.** Evolution of the ice surface at equidistant points up and downstream of the nunatak summit along the centreline (cf. Fig. 5c,d) in control runs with one nunatak with (a) the refined mesh (first presented in Fig. 5c), (b) the 5 km regular mesh, and (c) the 20 km regular mesh. Complementing Fig. 9, this figure shows that different estimates of surface elevation in the coarser mesh experiments also affect how elevation evolves as the ice thins, and therefore the resulting thinning rates.

its downstream counterpart. The rates of thinning, and consequently the timing of bedrock exposure, are dependent on the choice of the basal sliding coefficient. An increase of ca. 50 % in ice thinning was observed between the higher sliding and control experiments, as well as between the control and lower sliding experiments (not shown). This pattern is expected given the influence of basal sliding on the initial ice sheet geometry (Fig. S5), and highlights that the exposure lags between up and
downstream sides of a nunatak observed in the real world will be site dependent. The choice of minimum ice thickness used in our model (1 m) also influences the timing of exposure at a given point on the nunatak (Fig. S7). Although the lag times are still comparable to the range observed (2–16 kyr and 2–14 kyr, cf. Figs. 6b and S7b), a lower minimum thickness increased such lags, while a higher minimum thickness reduced them. The surroundings of nunataks are often crevassed, which results in a change in ice rheology. Hence, another test was carried out where the prescribed ice rheology becomes progressively
softer towards the nunataks by a factor of ten (Figs. S6 and S7). In this test, the most notable effect from the choice of ice rheology was a delay in the timing of exposure downstream of the nunatak relative to the control case, by 0.5–5 kyr, but the lag times were still of the same magnitude as the control case (i.e., between 2 and 10 kyr). While we use stabilisation of thinning upstream to determine whether the ice surface attained its minimum elevation before reaching the minimum thickness criterion, such stabilisation does not happen because of an equilibrium of the modelled upstream surface with the applied SMB. If the
upstream surface had attained equilibrium, stabilisation would have happened earlier when the imposed thinning was lower, which is the opposite of what was observed when comparing the three thinning scenarios. The fact that equidistant points up and downstream of the nunatak are in some cases not both exposed also implies that important changes in ice surface elevation might not be recorded upstream of a nunatak. The difference in time of exposure up and downstream is also higher in the experiments when ice flow is further constricted by the narrow glaciers (Fig. S11).
Idealised experiments have the advantage that certain interactions can be easily isolated and studied, but they also simplify certain phenomena that are observed in the real world, which could have influenced our results. For example, our setup is largely

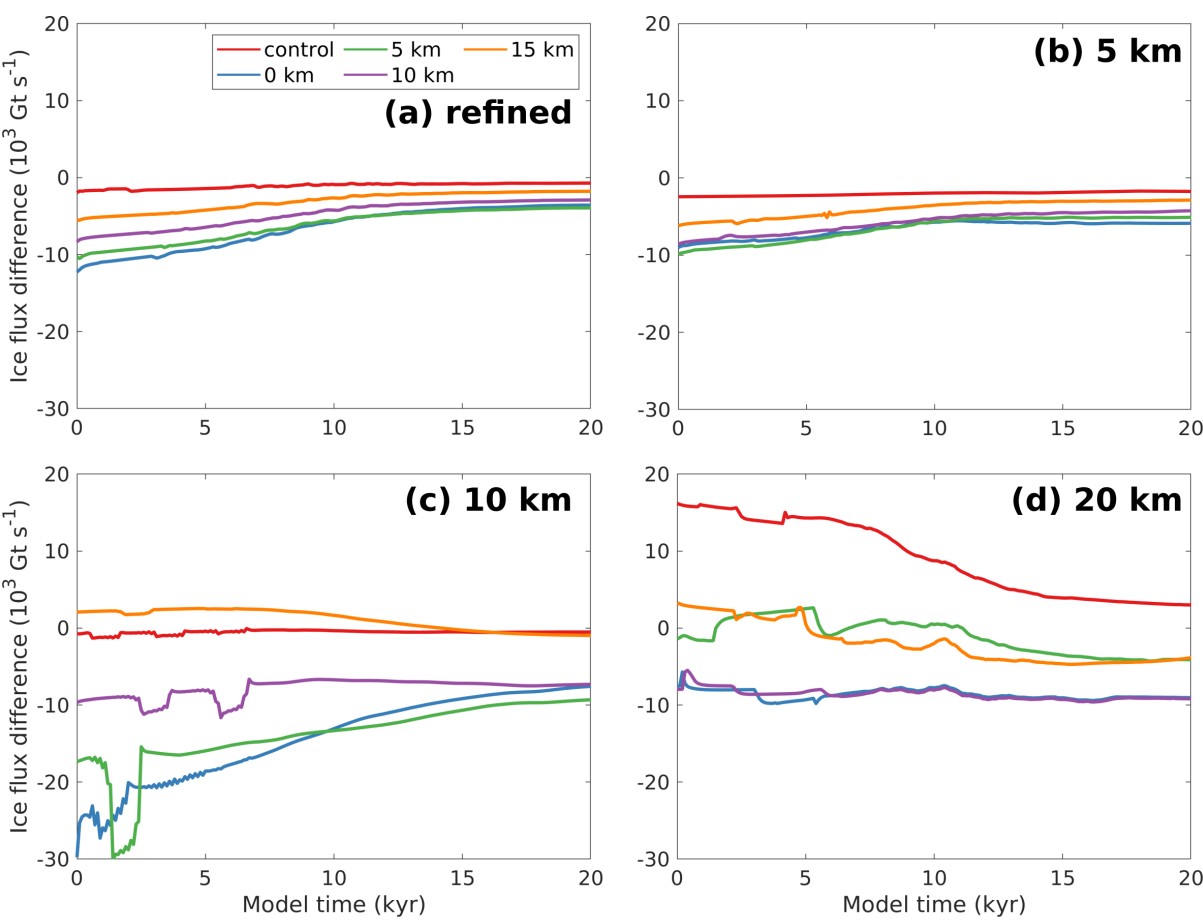

**Figure 11.** Difference between ice flux (in Gt s$^{-1}$) 20 km downstream and upstream from the nunatak summit (i.e., solid minus dashed lines in Fig. 8) over the entire simulation time for the experiments using (a) a refined mesh, and regular meshes of (b) 5 km, (c) 10 km and (d) 20 km horizontal resolution. This figure shows how the glacier width impacts ice advection downstream from the nunataks over time for all different model resolutions.

based on Antarctic settings, but the idealised SMB forcing deviates from what is more commonly observed in Antarctica in two ways. First, we prescribe ice thinning through surface melting, which is more characteristic of Greenland (e.g. Kjeldsen et al., 2015), while Antarctica's main source of mass loss is through dynamic thinning (Pritchard et al., 2009). Since mass 345 loss only occurs downstream of the nunataks and is highest at the downstream end of the model domain, we believe that the interaction between ice flow and nunataks, and the consequent ice surface elevation response, were not significantly impacted by how ice thinning was imposed. Second, the temporal perturbations to SMB are spatially uniform, which is indeed different from what was observed in areas of complex topography (e.g. Altnau et al., 2015). In regions where high mountains act as a barrier to the advection of moisture in the atmosphere, total variations in SMB are significantly lower inland of the 350 mountain ranges compared to coastal regions, which means that the surface elevation gradient found here and the lags in

surface exposure/stabilisation could be higher in non-idealised settings. A final set of sensitivity experiments (Fig. S8) shows that the ice surface elevation difference between locations equidistant up and downstream of a nunatak is more sensitive to the spatial gradient of prescribed SMB than to the absolute values of SMB. We do not consider glacial isostatic adjustment (GIA) in our experiments. Bedrock topography evolves through time due to changes in ice loading, which can influence the

sensitivity of the ice sheet to sea level forcing, potentially impacting the grounding line position (Whitehouse et al., 2019), and the cosmogenic production rate, possibly impacting the calculated exposure ages (Jones et al., 2019). In terms of patterns of exposure up and downstream of nunataks, the influence of differential isostatic bedrock uplift rates on each side of the nunatak are much smaller than the changes in ice surface elevation considered here, since bedrock elevation changes due to GIA happen at a spatial scale larger than the $\sim 60$ km observed in our experiments (e.g., Ivins and James, 2005). Despite the simplifications

mentioned above, important insights can be drawn from our idealised experiments regarding the impact of nunataks on ice-flow patterns, and how the differential response between the upstream and downstream surfaces introduce a bias on the timing of bedrock surface exposure around the nunatak. These effects are important for the interpretation of cosmogenic nuclide exposure ages and for comparing such ages with results from ice sheet models, and are discussed in more detail next.

## 4.2 Implications for the interpretation of past ice sheet reconstructions

The steepening of the ice surface around nunataks has important implications for the interpretation of in-situ constraints on past ice thickness changes from surface exposure dating. A commonly adopted practice is to assume that regional ice surface elevations are directly reflected by the absolute elevations of samples, but this is likely to yield inaccurate results for past ice sheet reconstructions. Our study demonstrates that this assumption would often yield an error of up to almost 400 m (see Fig. 5), which is of the same magnitude as many reported thickness-change estimates in regions of significant ice surface relief

(e.g. Ackert et al., 2007; Suganuma et al., 2014; Kawamata et al., 2020). A different practice, which allows for improved comparisons between sites, uses sample elevations relative to the modern ice surface elevation to infer past ice sheet thickness changes (e.g. Johnson et al., 2008; Jones et al., 2015). A key assumption in this approach is that the ice surface gradient and organisation of ice flow around a nunatak remained the same through time. This assumption contradicts our modelling results, which show increasing ice surface gradients around nunataks during ice sheet thinning. Our experiments further indicate

that when samples are taken upstream (downstream) of a nunatak, estimates of past regional ice surface elevation will be overestimated (underestimated). In directions transverse to ice flow, the ice surface is typically at a lower (higher) elevation than directly upstream (downstream) of the nunatak summit. The interpretation of thickness evolution is further complicated considering that the direction of ice flow could have changed as the ice thinned (e.g. Fogwill et al., 2014; Suganuma et al., 2014). This means that using the present-day ice surface as reference elevation could lead to misleading results.

An alternative practice for inferring ice thickness changes is to determine minimum and maximum estimates. In a recent study by Andersen et al. (2020), three estimates were provided to put bounds on ice thickness changes, recognizing that ice covering the sampled sites could have been sourced locally (from nearby higher terrain), regionally, or distally (the latter two referring to a thickening of the nearby ice stream Jutulstraumen). For minimum estimates of required thickening to reach the sampled elevation, two reference points on the present-day ice surface were determined. A "local" reference point was defined

by the lowest elevation in locally-sourced ice surrounding a nunatak. A "regional" reference point was defined by the major break-in-slope between the adjacent ice stream surface and more stagnant ice flanking the sampled nunatak (normally within 1–2 km from the nunatak). For a maximum estimate of thickening, the lowest point on the ice stream served as reference (lowest point along a 100 km profile perpendicular to ice flow in the ice stream and across the sampled site). We find that these estimates of minimum and maximum thickening are qualitatively comparable to our modeling results. Minimum estimates are similar to the difference between sample elevation and reference ice surface elevation for samples on the downstream side of nunatak summits. Conversely, maximum estimates could yield overestimated thickness changes of several hundreds of metres – in this sense it is comparable to samples from the upstream side of the nunatak summits in our experiments, when referenced to the lowest elevation present-day ice surfaces, usually downstream of the nunatak or along the ice stream.

Our modelling results can also be used to guide the collection of samples for cosmogenic dating. Typically, nunatak flanks are likely to provide the most accurate estimates of regional ice sheet thickness change, as these areas are least impacted by differences in ice surface steepening (cf. dashed coloured lines and dotted black line in Fig. 5a,b, and their respective locations in Fig. 3). Also, sampling at the nunatak flanks should diminish the ice surface gradient effect on exposure ages, while samples taken from the up or downstream sides would yield increased age differences (see Sect. 4.1). While nunatak flanks are commonly sampled, there has been no strong preference for such locations (Fig. 2). We assess the degree to which our model results are reflected in empirical data by analysing all current cosmogenic nuclide exposure ages from nunataks in Antarctica (Balco, 2021; Heyman, 2021) that indicate ice recession since the Last Glacial Maximum (last 21 kyr). Choosing this time interval minimises the effect of inherited concentrations from prior exposure, and confirms that samples taken upstream and downstream at the same elevation interval show significant age differences. This pattern exists regardless whether exposure ages from $^{10}$Be (which are most ubiquitous) or $^{14}$C (which are less susceptible to inherited concentrations) are considered (Fig. S12). Age gradients are not as pronounced for samples taken from nunatak flanks (Fig. S13). However, complex exposure histories and different sampling strategies among studies, which target specific thinning histories, introduce a spatial bias to the data, and to our knowledge, no sampling strategy has been designed to test for lags in exposure up and downstream of nunataks. Such experiments would aid the validation and interpretation of our findings. Regardless, we recommend that in addition to reporting cosmogenic nuclide concentrations, apparent ages, and uncertainties, the sample location relative to ice flow near the nunatak should be specified. This additional information would be crucial in the interpretation of the local ice sheet history, and would aid a domain-wide comparison of field constraints and modelling results.

The recommendations for sample collection based on our results also apply to subglacial bedrock locations that are targeted to test for past ice sheet collapse (e.g. Spector et al., 2019). In particular, sampled subglacial bedrock ridges on the downstream side of a nunatak may record past exposure indicative of a thinner-than-present ice sheet, but an equivalent subglacial ridge on the upstream side may record no such exposure. High-resolution ice flow modelling around a sampled nunatak is therefore necessary to understand how representative the sample location is of regional-scale ice loss. Irrespective of sampling strategy and application, it is important to keep in mind that not only modern ice flow direction should be considered, but also past variations in flow patterns. Local signs of palaeo ice flow (e.g. glacial striations) can help with such interpretations.

## 4.3 Implications for modelling ice flow in areas of large topographic relief

Model resolution plays a substantial role in how modelled ice flow interacts with nunataks. Regular-mesh experiments with resolutions typical for ice sheet models that simulate multi-millennial changes (5–20 km), show a more pronounced elevation gradient between the ice surface up and downstream of the nunataks than experiments with a variable mesh resolution. These resolutions do not properly capture the prescribed nunatak shape, patterns of ice flow or thinning rates, which are different in each experiment despite the same applied forcing. Palaeo ice sheet models are run at relatively coarse horizontal resolutions

(5–40 km; e.g. Golledge et al., 2012; Whitehouse et al., 2012; De Boer et al., 2014; Kingslake et al., 2018; Tigchelaar et al., 2018; Gomez et al., 2020) to keep computational times reasonable, and often use cosmogenic exposure dates as constraints (i.e. to define model parameters) or benchmarks (i.e. to assess the model's ability to reproduce the geological record). Experiments using a regular grid at these resolutions do not resolve site-specific ice surface elevation responses, which contributes to mismatches relative to reconstructions from cosmogenic exposure dating (Spector et al., 2019; Stutz et al., 2020; Johnson

et al., 2021). In our experiments, a grid-cell size smaller than the glacier width manages to capture the drainage effect to some degree. Still, a higher number of grid cells is needed to properly resolve ice flow through the narrow glaciers between nunataks and over large topographic relief, thus resolving the observed differences in deglaciation age up and downstream of nunatak summits. A better representation of ice flow diminishes the overestimation of the ice-surface elevation gradient. To overcome this limitation in palaeo ice sheet models, the model grid could be refined around nunataks so that it properly resolves

this pattern of ice flow. The use of adaptive meshes in ice sheet models (e.g. Berends et al., 2021), nested regional models, or downscaled setups, which take a lower-resolution ice sheet model state as boundary/initial conditions, could be potential solutions. Furthermore, due to their appropriate representation of regional ice flow patterns, higher resolution simulations could also help when designing sampling strategies and reconstructing regional thickness changes, thus diminishing the mismatch between modelled and reconstructed ice surface elevation. In summary, we advocate for a closer collaboration between ice

sheet modellers and field scientists. This would allow for common goals to be defined and for both disciplines to help each other overcome their respective inherent limitations.

## 5 Summary and conclusions

Ice flow in regions of complex terrain, where mountain ranges create steep ice-surface profiles, provide challenges for reconstructing and modelling past ice sheet changes. The choice of a reference present-day ice surface elevation to be used when

determining regional estimates of thickness change from surface exposure dating is not straightforward, and models struggle to match the available elevation reconstructions and their timing of surface exposure. In order to improve our understanding of ice flow over these regions of large topographic relief, we used an ice flow model that represented an idealised portion of an ice sheet. Five experiment ensembles were carried out in order to better understand how the ice surface responds to the presence of nunataks under thinning scenarios, and how the local response compares to the regional response. The first ensemble comprised

simulations where different degrees of ice thinning and different shapes (elongated transverse and parallel to ice flow) were tested for a single nunatak. The other four sets were performed to assess the interaction of three obstacles aligned transverse to

flow with the ice surface, and to which extent grid resolutions commonly employed by ice sheet models capture smaller-scale ice flow patterns between these obstacles.

Overall, we find that the interaction of ice flow with a nunatak results in a steepening of the ice surface, caused by increasing elevation upstream and lowering downstream. Locally, this ice surface mismatch results in an earlier bedrock exposure downstream, and a delayed exposure upstream during ice sheet thinning. At a regional scale, a single nunatak is able to impact the ice surface up to 30 km in both directions along the centre line, while a surface steepening is present transverse to ice flow over the entire extent to which the subglacial continuation of a nunatak stands out from the otherwise linearly sloping bedrock elevation. As a result of these steeper ice surface gradients, points at equal distances upstream and downstream of nunatak summits can differ as much as 14 kyr in surface exposure following ice sheet thinning. Although a compilation of cosmogenic nuclide ages indicates that samples taken upstream and downstream yield different ages, spatial biases in the sample distribution could have introduced biases in the resulting age distribution. A positive interference between closely spaced nunataks, or a more extensive nunatak perpendicular to flow can further increase the ice surface elevation gradient, while efficient drainage through glaciers formed between the nunataks is able to alleviate it. This mismatch and its consequences should be taken into account when sampling for surface exposure dating and when inferring past ice sheet thickness change, since they directly influence the interpretation of sample elevation relative to the regional ice surface.

We find that the current grid resolutions employed by ice sheet models cannot adequately resolve ice flow through nunatak ranges. The inability to capture smaller scale interactions between ice flow and bed topography results in an overestimated ice surface gradient across these obstacles, and the models miss important variations in the time of bed exposure up and downstream of nunatak summits. A resolution of 5 km can to some degree resolve a glacier width of 10 km or more, reducing the overestimation but not entirely solving the problem. Although such high-resolution simulations are currently too computationally expensive to be carried out at full ice-sheet scale, especially over millennia, accurate data-model comparisons with cosmogenic-nuclide dated surfaces require a proper regional refinement around the nunataks. This could be achieved with the aid of adaptive meshes, nested grids, or higher resolution regional models, which should perform better at reproducing the timing and magnitude of ice loss in regions of complex and large topographic relief. Better understanding the relationship between sample location and regional patterns in ice flow and ice surface elevation, combined with improved model simulations over sampled sites, will ultimately allow to account for potential biases in exposure ages and improve the comparison between in-situ data and ice sheet models. This should be achievable by a closer collaboration between ice sheet modellers and field scientists.

*Code availability.* The source code for Úa is available on https://github.com/GHilmarG/UaSource (Gudmundsson, 2020). All configuration files necessary to reproduce the experiments are available on https://github.com/martimmas/Nunatak_modelling.

*Author contributions.* JN and RSJ had the idea. MMB, JN, and RSJ developed the experiments, with input from APS and IR. MMB conducted and analysed the experiments, and JN provided the nunatak profiles for model validation. MMB wrote the manuscript with input from all authors.

*Competing interests.* Arjen Stroeven is editor of TC

*Acknowledgements.* This work is funded by the MAGIC-DML project. MAGIC-DML is a consortium supported by Stockholm University (Arjen Stroeven), the Norwegian Polar Institute/NARE (Ola Fredin), the US National Science Foundation (Nathaniel Lifton and Jonathan Harbor), the Swedish Research Council (Jonathan Harbor and Arjen Stroeven), and the German Research Foundation (DFG) Priority Programme 1158 "Antarctic Research" (Irina Rogozhina and Matthias Prange). RSJ was supported by a Junior Research Fellowship cofunded

between Durham University and the European Union. MMB would like to thank Hilmar Gudmundsson, and Jorge Bernales for providing insightful feedback on an earlier version of the model setup, and Henning Åkesson for providing comments on the manuscript. The computations and data handling were enabled by resources provided by the Swedish National Infrastructure for Computing (SNIC) at the National Supercomputer Centre (NSC) partially funded by the Swedish Research Council through grant agreement no. 2018-05973

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
