# Peer review of "Nunataks as barriers to ice flow: implications for palaeo ice-sheet reconstructions"

_The Cryosphere, 2021_

## Referee Comment (RC3)

This manuscript by Mas e Braga et al. explores the effect of nunataks on ice surface elevation during periods of ice thinning. This is important because the best available records of past ice sheet thinning are from cosmogenic exposure ages of glacial erratics collected on the slopes of nunataks. The findings of their idealized ice sheet model simulations suggest that samples collected upstream and downstream of a nunatak would produce different surface exposure ages (up to 14 kyr difference) due to an increase in ice surface elevation upstream and decrease in elevation downstream. Coarsening model resolution leads to an underestimation of this surface steepening effect. The study is well-written and presented, and should be of interest to readers of The Cryosphere. However, I hope the authors can address the following comments and questions. This review is divided into general and specific comments.

General comments:

1. Choice of model: The study addresses the issue of nunataks acting as a barrier to ice flow using an SSA model, with the authors correctly noting using a full Stokes model would be infeasible considering the long timescales involved (i.e. 20 kyr). I agree with Reviewers 1 and 2 that this warrants a robust discussion of model limitations given the 3D flow regime and steep gradients that would be present near nunataks. In addition, I think it would be useful for the authors to draw on the results of model intercomparison projects (e.g. MISMIP+, Cornford et al., 2020) to discuss whether other types of models would behave similarly or not. I'd note that SSA models are atypical for paleo-applications, with the majority of such studies using hybrid models and, more recently, higher order models, so a discussion of the model dependency of the findings in this study would be helpful.

2. Context of findings: The authors assert throughout the paper that "ice sheet models overestimate ice surface elevations and underestimate the pace of ice sheet melt contributing to sea level rise compared to empirical reconstructions." I find this to be an oversimplification for a number of reasons. There are many examples of ice sheet models that fit empirical constraints during glacial and interglacial periods (e.g. Whitehouse et al., 2012; Golledge et al., 2013; Goezler et al., 2016; Whitehouse et al., 2017; Clark et al., 2020), as well as the rate of mass loss during deglaciations and through the Holocene (e.g. Gomez et al., 2013; Lecavalier et al., 2014; Tigchelaar et al., 2018; Cuzzone et al., 2019; Briner et al., 2020; Albrecht et al., 2020a). While two studies are cited with records from the Transantarctic Mountains (Lines 72-73), even models analysed within those studies show similar rates of ice thinning as observed in the records. I agree with the authors that data-model mismatches occur in both Greenland and Antarctica, particularly with respect to the timing of regional ice thinning, though I'm not sure that the findings here definitively rule out other possible explanations, such as uncertainty in climate forcing (e.g. Lowry et al., 2019; Albrecht et al., 2020a), or uncertainty in the surface exposure ages themselves (e.g. Jones et al., 2019). It would be more accurate to simply point out that many previous paleo-ice sheet model studies, which use coarse resolution, do not resolve site-specific features and this could contribute to observed timing mismatches between surface exposure records and model simulations.

3. Influence of GIA: I am surprised that glacial isostatic adjustment was not considered in the simulations given the timescale. The calculation of surface exposure ages is highly dependent on the elevation, and correcting for GIA-driven elevation changes in non-trivial (Jones et al., 2019). Ice sheet models themselves are highly sensitive to GIA because it impacts bed elevation (e.g., Gomez et al., 2013; Kingslake et al., 2018; Colleoni et al., 2018; Albrecht et al., 2020b). While this study uses idealized experiments, in reality the basal topography

would evolve through time in response to changes in ice loading. The authors do provide a useful discussion on limitations in the simplified SMB forcing for the idealized experiments (Lines 320-334), but a similar discussion with respect to GIA is currently lacking.

Specific comments:

Line 16-17: See above general comment on context of findings.

Fig 2: Is there a scale for the slices? The colours correspond to upstream and downstream, correct? The point is to show that samples are taken in nearly all directions?

Line 136: citation? Does SMB only change in one direction?

Line 160: Any sensitivity tests for this value? 5-6% $K^{-1}$ is appropriate for EAIS (Frieler et al. 2015). WAIS is less straightforward (Fudge et al. 2018).

Fig 4: Could you make the inset larger? It is difficult to see the coloured circles that correspond to panels c and d.

No technical corrections noted.

References

Albrecht, T., Winkelmann, R., & Levermann, A. (2020a). Glacial-cycle simulations of the Antarctic Ice Sheet with the Parallel Ice Sheet Model (PISM)–Part 1: Boundary conditions and climatic forcing. *The Cryosphere*, *14*(2), 599-632.

Albrecht, T., Winkelmann, R., & Levermann, A. (2020b). Glacial-cycle simulations of the Antarctic Ice Sheet with the Parallel Ice Sheet Model (PISM)–Part 2: Parameter ensemble analysis. *The Cryosphere*, *14*(2), 633-656.

Briner, J. P., Cuzzone, J. K., Badgeley, J. A., Young, N. E., Steig, E. J., Morlighem, M., ... & Nowicki, S. (2020). Rate of mass loss from the Greenland Ice Sheet will exceed Holocene values this century. *Nature*, *586*(7827), 70-74.

Cornford, S. L., Seroussi, H., Asay-Davis, X. S., Gudmundsson, G. H., Arthern, R., Borstad, C., ... & Yu, H. (2020). Results of the third Marine Ice Sheet Model Intercomparison Project (MISMIP+). *The Cryosphere*, *14*(7), 2283-2301.

Clark, P. U., He, F., Golledge, N. R., Mitrovica, J. X., Dutton, A., Hoffman, J. S., & Dendy, S. (2020). Oceanic forcing of penultimate deglacial and last interglacial sea-level rise. *Nature*, *577*(7792), 660-664.

Colleoni, F., De Santis, L., Montoli, E., Olivo, E., Sorlien, C. C., Bart, P. J., ... & Prato, S. (2018). Past continental shelf evolution increased Antarctic ice sheet sensitivity to climatic conditions. *Scientific reports*, *8*(1), 1-12.

Cuzzone, J. K., Schlegel, N. J., Morlighem, M., Larour, E., Briner, J. P., Seroussi, H., & Caron, L. (2019). The impact of model resolution on the simulated Holocene retreat of the

southwestern Greenland ice sheet using the Ice Sheet System Model (ISSM). *The Cryosphere*, *13*(3), 879-893.

Frieler, K., Clark, P. U., He, F., Buizert, C., Reese, R., Ligtenberg, S. R., ... & Levermann, A. (2015). Consistent evidence of increasing Antarctic accumulation with warming. *Nature Climate Change*, *5*(4), 348-352.

Fudge, T. J., Markle, B. R., Cuffey, K. M., Buizert, C., Taylor, K. C., Steig, E. J., ... & Koutnik, M. (2016). Variable relationship between accumulation and temperature in West Antarctica for the past 31,000 years. *Geophysical Research Letters*, *43*(8), 3795-3803.

Goelzer, H., Huybrechts, P., Loutre, M. F., & Fichefet, T. (2016). Last Interglacial climate and sea-level evolution from a coupled ice sheet–climate model. *Climate of the Past*, *12*(12), 2195-2213.

Golledge, N. R., Levy, R. H., McKay, R. M., Fogwill, C. J., White, D. A., Graham, A. G., ... & Hall, B. L. (2013). Glaciology and geological signature of the Last Glacial Maximum Antarctic ice sheet. *Quaternary Science Reviews*, *78*, 225-247.

Gomez, N., Pollard, D., & Mitrovica, J. X. (2013). A 3-D coupled ice sheet–sea level model applied to Antarctica through the last 40 ky. *Earth and Planetary Science Letters*, *384*, 88-99.

Jones, R. S., Whitehouse, P. L., Bentley, M. J., Small, D., & Dalton, A. S. (2019). Impact of glacial isostatic adjustment on cosmogenic surface-exposure dating. *Quaternary Science Reviews*, *212*, 206-212.

Kingslake, J., Scherer, R. P., Albrecht, T., Coenen, J., Powell, R. D., Reese, R., ... & Whitehouse, P. L. (2018). Extensive retreat and re-advance of the West Antarctic Ice Sheet during the Holocene. *Nature*, *558*(7710), 430.

Lecavalier, B. S., Milne, G. A., Simpson, M. J., Wake, L., Huybrechts, P., Tarasov, L., ... & Larsen, N. K. (2014). A model of Greenland ice sheet deglaciation constrained by observations of relative sea level and ice extent. *Quaternary Science Reviews*, *102*, 54-84.

Lowry, D. P., Golledge, N. R., Bertler, N. A., Jones, R. S., & McKay, R. (2019). Deglacial grounding-line retreat in the Ross Embayment, Antarctica, controlled by ocean and atmosphere forcing. *Science advances*, *5*(8), eaav8754.

Tigchelaar, M., Timmermann, A., Pollard, D., Friedrich, T., & Heinemann, M. (2018). Local insolation changes enhance Antarctic interglacials: Insights from an 800,000-year ice sheet simulation with transient climate forcing. *Earth and Planetary Science Letters*, *495*, 69-78.

Whitehouse, P. L., Bentley, M. J., & Le Brocq, A. M. (2012). A deglacial model for Antarctica: geological constraints and glaciological modelling as a basis for a new model of Antarctic glacial isostatic adjustment. *Quaternary Science Reviews*, *32*, 1-24.

Whitehouse, P. L., Bentley, M. J., Vieli, A., Jamieson, S. S., Hein, A. S., & Sugden, D. E. (2017). Controls on last glacial maximum ice extent in the Weddell Sea embayment, Antarctica. *Journal of Geophysical Research: Earth Surface*, *122*(1), 371-397.

---

## Author Response (AR1)

Dear Editor,

We would like to thank you and the reviewers for the time invested in revising our manuscript. Below you will find the response letters to all three reviewers.

We hope that you find our manuscript suitable for publishing in The Cryosphere.

Best regards,

Martim Mas e Braga

Dear Anonymous Reviewer 1, Dear Editor,

We would like to thank the reviewer for the insightful comments and good feedback provided on our manuscript. Below we provide answers to each comment. The original comments are numbered in **red**, and written in **bold**. Our answers are provided in black, and the manuscript text including the proposed changes is provided in *"quoted italics"*. For an easier comparison between this response letter and the original manuscript, we refer to the figures by their original numbering, despite new figures being presented here (and proposed for inclusion). New figures are presented and referenced in this letter as "new figure", and their numbering will be adjusted in the revised manuscript.

**Major issues:**

**1. This study uses an SSA model with a minimum ice thickness to simulate shallow ice flow around high-relief obstacles. In principle, I can see how this model is appropriate, but it is a fairly non-standard use of SSA. For one, it is unclear to what extent SSA applies for ice that changes from kilometer-scale thickness to meter scale thickness. Further, even if the exposure threshold you use is above the minimum imposed modeled thickness, the use of a minimum in of itself will potentially affect how ice flow occurs near the ice margin on the Nunatak (i.e. constant thickness at the minimum over space vs. gradually decreasing thickness over space). There are also missing details regarding the stress and flux boundary conditions at these sorts of margins. At a minimum, the model descriptions needs more details and justification for why we should believe SSA in such a situation (as opposed to, say, Full Stokes). It may also help to investigate how the results (particular in terms of simulated exposure timing) depend on the choices of minimum model ice thickness and the boundary conditions therein.**

We agree it is a non-standard use of SSA. Its use is justified by the lower computational cost of this approximation compared to a full-Stokes model, the target settings we aim to represent, and the model's dynamic adaptive meshing capability. We also agree that full-Stokes models would yield more accurate results, and had highlighted this in our original manuscript (L87). Based on the comments and suggestions above and from reviewers 2 and 3, we have expanded Sect. 2 to better justify the validity of the SSA approximation, clarify the boundary conditions used in our model, as well as its limitations compared to a full-Stokes model.

[revised manuscript text omitted]

***New Supplementary Figure:*** *Spatial distribution of the rheology factor A for the 'crevassed' experiment (*[ref. to new figure below]*). The value for A away from the nunatak is that used in all other experiments (log$_{10}$(A) ~ -8.5).*

[Figure]

***New Supplementary Figure:*** *As in Fig. 5 of the main text, but for minimum ice thickness experiments where the minimum thicknesses allowed in the model are 0.5 and 5 m, and a*

*"crevassed" experiment, where the ice is softer around the nunatak compared to the rest of the model domain (ref to Fig, above).*

**References not previously in the manuscript:**

Cornford, S.L., et al. (2020). Results of the third Marine Ice Sheet Model Intercomparison Project (MISMIP+). *The Cryosphere*, *14*, 2283-2301.

Durand, G., Gagliardini, O., De Fleurian, B., Zwinger, T., & Le Meur, E. (2009). Marine ice sheet dynamics: Hysteresis and neutral equilibrium. *Journal of Geophysical Research: Earth Surface*, *114, F03009*.

Fowler, A. C., & Larson, D. A. (1978). On the flow of polythermal glaciers-I. Model and preliminary analysis. *Proceedings of the Royal Society of London. A. Mathematical and Physical Sciences*, *363*(1713), 217-242.

Ito, K., & Kunisch, K. (2008). *Lagrange multiplier approach to variational problems and applications*. Society for Industrial and Applied Mathematics.

Wirbel, A., & Jarosch, A. H. (2020). Inequality-constrained free-surface evolution in a full Stokes ice flow model *(evolve_glacier v1.1)*. *Geoscientific Model Development*, *13*, 6425-6445.

**2. I come away from reading this study mostly convinced that there are potential issues with the way cosmogenic exposure ages are interpreted. However, I would also have liked to see a bit more concrete solutions/suggestions to solve this problem. Is the answer that every time someone measures a cosmogenic exposure ages, they will have to do detailed high-resolution modeling of ice flow in the region from that time period to place the specific measurements in context? This seems like quite a lot of work. Two less onerous possibilities (only one of which is currently mentioned in the manuscript) intrigue me: (1) can already-measured exposure ages be corrected using the results of this study? What would be necessary to do so? Does one need multiple ages from different parts of the same Nunatak to do such a correction? (2) It seems like you suggest in the discussion that sampling Nunatak at locations that are perpendicular to flow (i.e. on the "sides") would mitigate the age bias somewhat. However, most of your discussion of your experiments was focused on the difference between upstream and downstream. Overall, I think the focus of the discussion needs to be shifted towards providing potential ways forward on a solution, particularly for those who may not have the capability to do detailed simulations in support for exposure ages.**

We are reluctant to provide absolute values for corrections because regional settings such as mass balance, bedrock slope, and ice stream/glacier widths exert significant influence in the ice change history. Nevertheless, we have rewritten some parts of the discussion based on the proposed suggestions (detailed below), highlighting the recommendations that were already provided, and also the ways we believe the issues raised in our study can be best addressed:

- Sampling at nunatak flanks. Also addressing point 34, we have further included a comparison of cosmogenic-nuclide ages taken at the nunatak flanks (see new supplementary figure below), which complements Fig. S9, and refer to it in the same sentence:

*"We assess the degree to which our model results are reflected in empirical data by analysing all current cosmogenic nuclide exposure ages from nunataks in Antarctica (Balco, 2021; Heyman, 2021) that indicate ice recession since the Last Glacial Maximum (last 21 kyr). Choosing this time interval minimises the effect of inherited concentrations from prior exposure, and confirms that samples taken upstream and downstream at the same elevation interval show significant age differences. This pattern exists regardless whether exposure ages from $^{10}$Be ( which are most ubiquitous) or $^{14}$C (which are less susceptible to inherited concentrations) are considered (Fig. S9). Age gradients are not as pronounced for samples taken from nunatak flanks ([ref. to new supplementary figure below])."*

- Sampling at areas where model predictions of lags can be tested would be useful for validating our results. For unpublished already-collected samples, we highlight that their position relative to flow should be reported (originally mentioned in L371 – 375). We have expanded on this sentence to draw the attention to the fact that the sample placement relative to ice flow should be considered when interpreting the data:

*"Such experiments would aid the validation and interpretation of our findings. Regardless, we recommend that in addition to reporting cosmogenic nuclide concentrations, apparent ages, and uncertainties, the sample location relative to ice flow near the nunatak should be specified. This additional information would be crucial in the interpretation of the local ice sheet history, and would aid a domain-wide comparison of field constraints and modelling results."*

- Also, we see our paper as a call for closer collaboration between field scientists and ice sheet modellers in designing sampling strategies and interpreting results. We present this last point explicitly towards the close of our discussion section:

*"In summary, we advocate for a closer collaboration between ice sheet modellers and field scientists. This would allow for common goals to be defined and for both disciplines to help each other overcome their respective inherent limitations."*

And in the end of our conclusions:

*"This should be achievable by a closer collaboration between ice sheet modellers and field scientists."*

[Figure]

***New Supplementary Figure.*** *Sample location relative to flow and age distribution for $^{10}$Be (upper row) and $^{14}$C (lower row) as in Fig. S9, but considering only the samples taken at the nunatak flanks.*

**3. I found the text of section 3 very dense with descriptions of results. Rather than re-describe a lot details that are in the figures, making more effort to synthesize results and describe their physical meaning would be welcome. Also, generally shortening section 3 would help as well.**

Wherever possible in section 3, we have made changes to be more concise in reporting our results, highlighting their physical meaning where they were not clear. However, we did not focus on them, as it would cause significant overlap with the discussion section, where we believe such focus is more fitting, and where we contextualise our results.

**Minor issues:**

**4. Throughout: new results should be described using the present tense, whereas prior studies should be described using past tense. There is some inconsistency throughout on this.**

We have revised the manuscript and changed accordingly.

**5. Line 15: while rapid ice flow through outlet glaciers...**

we used "efficient ice drainage" to highlight the fact that a constricted flow also contributes to the formation of a steeper ice surface elevation gradient, i.e., it is not necessarily the ice flow velocity that controls it.

**6. Line 15: what do you mean by "alleviated the differential response" - this is confusing**

we have rephrased the end of the sentence, which now reads:

*"A nunatak elongated transverse to ice flow is able to increase ice retention and therefore impose steeper ice surface gradients, while efficient ice drainage through outlet glaciers produces gentler gradients."*

**7. Line 23-43: These two paragraphs are quite general and can probably be shortened to one paragraphs**

We have merged both paragraphs, shortening the content by 25% without the loss of important information. We believe it is important to acknowledge the different time scales at which ice sheet modellers work, highlight the most recent improvements in ice sheet modelling, and show that a good representation of complex bedrock topography remains a knowledge gap. The first paragraph of the manuscript now reads:

*"Ongoing changes in climate are already causing significant mass loss and ice-margin retreat of both the Antarctic and Greenland ice sheets (Garbe et al., 2020; King et al., 2020). Near-future (2100 CE) projections of sea level rise point to ocean thermal expansion as the main cause (Oppenheimer et al., 2019), but over multi-centennial timescales, the sea level contribution from Antarctica is expected to become dominant (Pattyn and Morlighem, 2020). Numerical ice sheet modelling efforts are aimed at reducing uncertainty by better understanding the processes that lead to sea level rise, focusing on both shorter (Goelzer et al., 2020; Seroussi et al., 2020), and longer timescales (Pollard and DeConto, 2009; Albrecht et al., 2020). Recent efforts include improvements in key model components such as grounding line dynamics (e.g. Gladstone et al., 2017; Seroussi and Morlighem, 2018), coupling to solid Earth and sea level models (e.g. Gomez et al., 2020), and improved treatment of ice-ocean interaction processes (e.g. Reese et al., 2018; Kreuzer et al., 2020). The importance of bedrock topography (Morlighem et al., 2020) and grid resolution (Durand et al., 2011) have been acknowledged previously, and studied particularly for marginal regions of the ice sheet (e.g. Sun et al., 2014; Robel et al., 2016; Favier et al., 2016). Spatial variations in bedrock topography, and the resulting basal and lateral drag exerted at the ice-bedrock interface for different spatial scales, can slow down or even stabilise grounding line retreat (Jamieson et al., 2012, 2014; Åkesson et al., 2018; Jones et al., 2021; Robel et al., In Review). Regions near the ice margin with large subglacial topographic relief, such as the overridden mountain ranges that fringe the glaciated cratons of Greenland and East Antarctica*

*(Howat et al., 2014; Burton-Johnson et al., 2016), therefore require suitable consideration when evaluating ice loss beyond this century."*

**8. Line 27: ice loss from marine-terminating outlet glaciers**

We have changed it accordingly.

**9. Line 40: drag is exerted on fjords?**

*This statement has been removed after the suggestions from comment 7*

**10. Line 56: setting where rock samples are acquired**

We have added "rock samples" accordingly.

**11. Line 69: that there is no systematic approach to selecting the sampling**

We have changed accordingly.

**12.** The following comments are grouped together for simplicity:

- **Figure 2: is the coloring of the bars necessary? The information seems redundant with the radial axis.**

- **Figure 2: later in the discussion you seemingly suggest that you might also have information on ages for each of these samples. If so, it would seem to make sense (or at least be interesting) to incorporate this extra data in this figure. Even if there is sampling bias, it would be interesting to show if there was a systematic difference in exposure ages in upstream and downstream samples. However, if this data is not available, thats not an issue.**

We have removed the colouring, and added a dashed line to help identify the threshold for considering a sample to be upstream or downstream. This was also done to the supplementary figures, for consistency. We have also added two panels to Fig. 2, showing the kernel density plots for $^{10}$Be and $^{14}$C ages considering all samples, as suggested:

[Figure]

**Figure 2.** (a) Polar histogram showing the location of cosmogenic $^{10}$Be and $^{14}$C samples from boulders in Antarctica (Heyman, 2021; Balco, 2021) with ages younger than the Last Glacial Maximum, relative to the nearest nunatak summit and its adjacent ice flow direction (n = 191; sample duplicates were excluded). The difference in direction was computed between a sample position relative to the nunatak summit (identified in BedMachine-Antarctica; Morlighem et al., 2020) and ice flow (Mouginot et al., 2012) near the nunatak summit. Summits were identified through a morphological feature map (Wood, 1996; see supplementary material). The area of each slice is proportional to the number of samples within that category, and each category spans a 15º arc. In this figure, 0º (180º) implies that the sample was taken downstream (upstream), directly aligned with the ice flow *direction. (b,c) kernel density function of $^{10}$Be (b) and $^{14}$C (c) apparent exposure ages from samples shown in panel (a). Dashed lines show the median age, and shading shows the uncertainty interval based on the median uncertainty of their respective ages*.

**13. Line 78: To perform these tests**

We have changed accordingly.

**14. Line 104: how is the ice front simulated?**

The ice front is treated as an equilibrium between the pressure exerted by the ice and the ocean (set at 0 m, as shown by the dashed line in Fig. 3b). The ice front is considered to retreat when the thickness reaches the prescribed minimum thickness, for which the active-set method is used. This is now included in the manuscript, see answer to comment 1.

**15. Line 118: I felt at this point like it would be helpful to have a figure just showing the bed geometries used in the simulations because it was hard to visualize from the text alone**

We have added a new figure showing bed elevation for both nunatak shapes. We added the profile lines and points used in Fig. 4, and removed its inset plots. We believe this change has made Fig. 4 cleaner, and addresses the issue raised in comment 25 regarding the meaning of the different lines.

[Figure]

**New Figure.** Bedrock elevation (m a.s.l.) used in the experiments where the nunatak is placed (a) transverse and (b) parallel to ice flow. Ice flow in this figure is from left to right. Transect lines show the position of the transects presented in Fig. 4a,b, and coloured circles show the locations for the ice surface evolution analysis presented in Fig. 4c,d following the same colours as the lines therein.

[Figure]

**Figure 4.** Surface elevations after 0, 10, and 20 kyr for the 'thw' experiment with a nunatak elongated (a) transverse and (b) parallel to ice flow at 0, 10, and 15 km from the centreline; dark grey lines show the respective ice surface elevation for experiments without a nunatak; (c,d) evolution of the ice surface elevation difference between six pairs of equidistant points up and downstream from the nunatak, along the centre line for the experiments showcased in (a,b) respectively. In short, this figure shows to which extent a single nunatak is able to influence ice surface elevation in space (along and perpendicular to flow) and through time, how ice surface elevation is linked to a nunatak's subglacial topography, and how the ice surface elevation mismatch evolves differently depending on the distance from the nunatak.

**16. Line 121: idealized nunatak dimensions**

We have changed accordingly.

**17. Line 122: are constrained by 33 real nunatak topographic profiles from Antarctica**
We have changed accordingly.

**18. Line 128: is there any expectation that your results are dependent on the sliding law, or coefficient chosen? Would more rapid sliding give a drastically different result?**
The fact that the sliding coefficient used was chosen to match that of an inversion, which also used Weertman's sliding law, prevents us from reliably choosing an equivalent coefficient for a different sliding law. However, the greatest differences between Coulomb-limited and Weertman-type laws occur in the vicinity of the grounding line. Given our mesh refinement over such regions, these differences are expected to be small (see Gladstone et al., 2017). This is now mentioned in our updated Sect. 2 (see response to comment 1).

We have also provided a comparison of total ice thinning, as opposed to time of exposure, for the different-sliding 'thw' scenario experiments, for which the initial states were originally presented in the supplement. The choice for ice thinning is motivated by the fact that their initial ice surface elevation is significantly different from one another. We mention it in the discussion (Sect. 4.1, second paragraph; see response to comment 1).

**19. Equation 2: is Lx = L*x or L_x?**

It is $L_x$. Thank you for spotting this typo.

**20. Line 145-146: confusing sentence**

We have slightly rephrased it as:

*"The average ice surface slope between the ice divide and the grounding line (which after spin up was located at x = 136 km; Fig. 3b), is ~1.3 %."*

**21. Section 2.5: here and elsewhere you say that typical paleo-ice sheet mode configurations are >5 km. For ice sheet wide simulations, sure, this is true, but I think for individual catchments, there are plenty of people doing ~1 km and higher resolution simulations, even for paleo simulations. Its pretty well within the realm of current SSA models to simulate such resolution on a spatially limited basis (even without adaptive meshing or other sophisticated techniques) for 10's of kyr.**

That is correct, thanks for highlighting this point. We meant indeed ice sheet-wide simulations, and have clarified it in the sentence:

*"In a final series of sensitivity experiments, we assess how well regular-spaced grids of coarser resolutions typically used in continental-scale ice sheet models (5, 10, and 20 km) resolve the ice surface elevation pattern around nunataks compared to the solution using an unstructured, locally refined mesh."*

**22.** **Line 190-193: I'm a bit confused because the ice intersects "above" the lowest ice surface elevation both upstream and downstream. Either this is a mistake, or more explanation is needed because it is counterintuitive that ice thickness would be higher downstream.**

Based on this comment and comment 3, we have rewritten the sentence to be more concise:

*"At the start of the thinning experiments where the nunatak is elongated transverse to flow, the effect of the nunatak on the ice surface elevation is seen up to 30 km away from its summit along flow, and 15 km perpendicular to flow. The ice surface directly upstream of the nunatak is 360 m above the general ice sheet surface (considered to be the elevation 15 km away from the centreline), while downstream the lowest ice surface elevation (located 3 km downstream of the summit) is 100 m lower than the general ice surface elevation."*

We have also provided a brief discussion regarding possible explanations for this phenomenon in the end of the first paragraph of Sect. 4.1:

*"A relative increase in the ice surface elevation was observed immediately downstream of the nunatak, while the lowest elevation attained was located ~3 km downstream of the nunatak summit at the start of the simulations. Although a similar effect can be observed on the lee side of isolated nunataks in Dronning Maud Land, MacRobertson Land, and the Transantarctic Mountains (Howat et al., 2019, Andersen et al., 2020), we cannot rule out that this a numerical effect, caused by artificial advection of ice downstream from the regions where the minimum thickness constraint is violated and thus numerically modified (see Jarosch et al., 2013)."*

**23. Line 199-201: confusing sentence**
We have rephrased the sentence, which now reads:

*"In both cases, changes in ice surface elevation perpendicular to ice flow, caused by the nunatak presence, stand out from the general ice surface elevation for the entire extent of the subglacial nunatak obstruction (cf. Figs. 4a,b, and [ref. to new figure showing the bed elevation]a,b))."*

**24. Line 213: this connection between slope and exposure is not very clear. Could be described more**

We have rewritten the mentioned sentences, also considering comment 3. This part now reads:

*"Both ice thinning experiments detailed in Fig. 4a,b reveal an overall steepening of the ice surface gradient during the 20 kyr thinning period (profiles from red to green). The evolution of this steepening is illustrated in Fig. 4c,d, where points equidistant from the nunatak summit (upstream and downstream) show ice surface elevation differences that increase through time. For equidistant locations closer to the nunatak summit (< 2.5 km and < 7.5 km for Figs. 4c and d, respectively) the increase in ice surface steepening (i.e., in elevation difference) is disrupted when the downstream location becomes exposed, since its elevation no longer changes while the equivalent point upstream is still ice covered and thinning."*

**25. Figure 4: it is not clear what all the different lines meaning**
We have removed the insets, and put the lines in the bed elevation figure, requested in comment 15.

**26. Figure 4: why does it appear like the "idealized" nunatak (brown in panels as and b) is bumpy? I'm guessing a plotting artifact, but its a bit distracting**
It was indeed a plotting artifact, and we thank the reviewer for spotting that. We believe we have improved it in the updated plot, as shown in our answer to comment 15, where we presented the changes to Figure 4.

**27. Figure 6: difference with respect to what?**
Figure 6 shows the difference between the three-nunatak experiments with respect to the one-nunatak control case, shown in panel (e). We have rephrased the caption to clarify this:

*"(a-d) Difference in surface elevation (m) between each three-nunatak 'thw' experiment of varying glacier widths (0, 5, 10, 15 km, respectively) with respect to the one-nunatak 'thw' control run (shown in panel e for reference), after 20 kyr of simulation"*

**28. Line 287: with-width**
Typo fixed accordingly.

**29. Line 294: response to what?**
To the nunataks. We have rephrased the sentence for clarity:

*"Our modelling experiments demonstrate that the magnitude of the ice surface elevation response due to the presence of nunataks is proportional to their ability to obstruct or constrict ice flow."*

**30. Line 299: In their fjord experiment, Frank**

We have changed accordingly.

**31. Line 309: between ice thickness on the up and downstream sides o the nunatak increases...**

We have changed accordingly.

**32. Line 327-328: would this add or subtract to the effect from flow around the nunatak? Could the whole exposure age discrepancy be compensated by this, or made even more severe? Seems like its worthy of a test in your model.**

We have performed two additional simulations, where the SMB profile is smoother at the upstream side of the domain, and decays rapidly downstream, through the following equation (cf. Eq. 2 of the main text):

$SMB_0 - (SMB_0 - SMB_e) * \tan(|x|/L_x) * \sin^2(|x|/L_x) + b(t)$

We prescribe two different values for $SMB_0$, -0.92 ma$^{-1}$ and -0.78 ma$^{-1}$, and call the respective experiments 'margin1' and 'margin2'. While 'margin1' ensures that the integrated SMB at t=0 kyr is the same as in the 'thw' scenario (referred here as 'control'), 'margin2' is designed so that the SMB upstream is close to zero. The new figure below shows how ice thinning is impacted by this different SMB curve, as in Fig. 4a of the manuscript. Overall, the smoother profile at the nunatak vicinity caused the surface elevation differences up and downstream to be smaller than in the 'control' run, although the general surface elevation is lower. Thus, the differences in ice surface elevation seem to be more sensitive to the gradient in SMB rather than the absolute values themselves.

[Figure]

**New Supplementary Figure.** (a) different surface mass balance (SMB) profiles analysed. Solid lines show the profile at the start of the simulations (t = 0 kyr), while dashed lines show the profile at the end (t = 20 kyr). The 'thw' profile (see Fig. 3a of the main text) is shown here as 'control'. (b – d) surface elevation profiles after 0, 10, and 20 kyr for each experiment referred in panel (a), as in fig. 4a of the main text. Grey lines show the same "no nunatak" experiment as in Fig. 4a for an easier comparison between all profiles.

We have added the above information to the supplementary material, and refer to it in the main text, adding to the SMB discussion highlighted by the reviewer (Sect. 4.1, last paragraph), as follows:

*"A final set of sensitivity experiments ([ref. to* new supplementary figure shown above]) *shows that the ice surface elevation difference between locations equidistant up and downstream of a nunatak is more sensitive to the spatial gradient of prescribed SMB than to the absolute values of SMB."*

**33. Line 338: where does the 400 m figure come from?**
It comes from the ice surface elevation difference at the ice-bedrock interface, and the general ice surface unaffected by the nunatak. We have referred to Fig. 4 to make it clearer.

**34. Line 361-364: I referenced this above, but this point seems very important and could do with more references to results, since you should be able to show this in your simulations**

We have added a reference to Fig. 4, where it is possible to compare the patterns of ice surface elevation to the sentence in question:

*"Typically, nunatak flanks are likely to provide the most accurate estimates of regional ice sheet thickness change, as these areas are least impacted by differences in ice surface steepening (cf. dashed coloured lines and dotted black line in Fig. 4a,b, and their respective locations in Fig.* [ref. to new figure showing the bed elevation]*)"*

We believe that this, combined with the other changes listed under our response to comment 2, have addressed this remark.

We thank the reviewer for the thorough and insightful review of our manuscript, and look forward to the opportunity to present a significantly updated manuscript based on the revisions outlined above.

Dear Anonymous Reviewer 2, Dear Editor,

We would like to thank the reviewer for the insightful comments and good feedback provided on our manuscript. Below we provide answers to each comment. The original comments are numbered in **red**, and written in **bold**. Our answers are provided in black, and the manuscript text including the proposed changes is provided in "*quoted italics*". For an easier comparison between this response letter and the original manuscript, we refer to the figures by their original numbering, despite new figures being presented here (and proposed for inclusion). New figures are presented and referenced in this letter as "new figure", and their numbering will be adjusted in the revised manuscript.

**(1) I have some doubts whether the shelfy-stream approximation (SStA) is appropriate for the tackled problem. SStA is SSA (shallow-shelf approximation) with basal drag, thus it assumes plug flow and a shallow geometry without sharp gradients. Especially when zooming in high resolution to the immediate vicinity of nunataks, these conditions are certainly violated, as the flow regime is strongly 3D and steep surface gradients occur. I think that a proper simulation of this behaviour requires nothing less than full-Stokes dynamics. There is probably not much to be done about the shortcoming at this stage, but at the minimum, the limitation should be discussed very openly and clearly.**

**BTW, which boundary condition is applied at the nunataks? I suppose they are treated like the ice base?**

We agree with the reviewer that a Full Stokes model would yield more accurate results, and explicitly mentioned that in the original manuscript (L87). Nevertheless, based on the comments above, as well as those provided by reviewers 1 and 3, we have expanded Sect. 2 where we present the used model's capabilities, highlighting its limitations as well as its features that also improve the representation of ice flow over our regions of interest. We also provide more information about how the boundary between ice and nunatak is handled in the model:

[revised manuscript text omitted]

Ito, K., & Kunisch, K. (2008). *Lagrange multiplier approach to variational problems and applications*. Society for Industrial and Applied Mathematics.

Wirbel, A., & Jarosch, A. H. (2020). Inequality-constrained free-surface evolution in a full Stokes ice flow model *(evolve_glacier v1.1)*. *Geoscientific Model Development*, *13*, 6425-6445.

**(2) What I am also missing is a discussion about the potential impact of crevassing in the vicinity of nunataks. I would expect that crevasses often occur around such flow obstacles, with the consequence that the ice is effectively softened and the large-scale flow less disturbed than under the assumption of undamaged ice that has to find its way around the obstacle. This effect has the potential to alter/weaken the described influence on the ice surface significantly. At least, a qualitative discussion about it deems appropriate. Even better would be to investigate the effect by running some tests with an assumed softening around the nunataks.**

The reviewer points out a very important aspect, which we overlooked. Hence, we now include a test under the same conditions as the 'thw' experiment with one nunatak, but considering a progressively softer ice (i.e., up to the equivalent of a temperature of -5 °C) as the ice gets thinner towards the nunatak summit. We added the spatial distribution of the rheology factor, and a comparison of how the time of exposure changes to the supplement, complementing the suggestion from Reviewer 1 (comment 1) about the sensitivity of the timing of exposure to our choice of minimum thickness. We refer to these tests when discussing our results (Sect 4.1, paragraph 2):

*"The ice surface steepening and consequent mismatch between the up and downstream sides of the nunatak increases as the ice thins, until the downstream side becomes exposed. Exposure happens earlier downstream, as expected due to lower ice surface elevation, and an equidistant point upstream becomes exposed (or has its thinning stabilised) up to 14 kyr later than its downstream counterpart. The rates of thinning, and consequently the timing of bedrock exposure, are dependent on the choice of the basal sliding coefficient. An increase of ca. 50 % in ice thinning was observed between the higher sliding and control experiments, as well as between the control and lower sliding experiments (not shown). This pattern is expected given the influence of basal sliding on the initial ice sheet geometry (Fig. S5), and highlights that the exposure lags between up and downstream sides of a nunatak observed in the real world will be site dependent. The choice of minimum ice thickness used in our model (1 m) also influences the timing of exposure at a given point on the nunatak ([ref. to the new supplementary figure below]). Although the lag times are still comparable to the range observed (2 – 16 kyr and 2 – 14 kyr, cf. Figs. 5b and [ref. to the new supplementary figure below]b), a lower minimum thickness increased such lags, while a higher minimum thickness reduced them. The surroundings of nunataks are often crevassed, which results in a change in ice rheology. Hence, another test was carried out where the prescribed ice rheology becomes progressively softer towards  the nunataks by a factor of ten ([ref. to the new supplementary figures below]). In this test, the most notable effect from the choice of ice rheology was a delay in the timing of exposure downstream of the nunatak relative to the control case, by 0.5-5 kyr, but the lag times were still of the same magnitude as the control case (i.e., between 2 and 10 kyr). While we use* stabilisation of thinning upstream to determine…"

[Figure]

**New Supplementary Figure:** *Spatial distribution of the rheology factor A for the 'crevassed' experiment ([ref. to new figure below]). The value for A away from the nunatak is that used in all other experiments ($\log_{10}(A) \sim -8.5$).*

[Figure]

 **New New Supplementary Figure:** *As in Fig. 5 of the main text, but for minimum ice thickness experiments where the minimum thicknesses allowed in the model are 0.5 and 5 m, and a "crevassed" experiment, where the ice is softer around the nunatak compared to the rest of the model domain (ref to Fig, above).*

We thank the reviewer for the thorough and insightful review of our manuscript, and look forward to the opportunity to present a significantly updated manuscript based on the revisions outlined above.

Dear Anonymous Reviewer 3, Dear Editor,

We would like to thank the reviewer for the insightful comments and good feedback provided on our manuscript. Below we provide answers to each comment. The original comments are numbered in **red**, and written in **bold**. Our answers are provided in black, and the manuscript text including the proposed changes is provided in *"quoted italics"*. For an easier comparison between this response letter and the original manuscript, we refer to the figures by their original numbering, despite a new figure being presented here (and proposed for inclusion). The new figures is presented and referenced in this letter as "new figure", but its numbering will be adjusted in the revised manuscript.

**General comments:**

**1. Choice of model: The study addresses the issue of nunataks acting as a barrier to ice flow using an SSA model, with the authors correctly noting using a full Stokes model would be infeasible considering the long timescales involved (i.e. 20 kyr). I agree with Reviewers 1 and 2 that this warrants a robust discussion of model limitations given the 3D flow regime and steep gradients that would be present near nunataks. In addition, I think it would be useful for the authors to draw on the results of model intercomparison projects (e.g. MISMIP+, Cornford et al., 2020) to discuss whether other types of models would behave similarly or not. I'd note that SSA models are atypical for paleo-applications, with the majority of such studies using hybrid models and, more recently, higher order models, so a discussion of the model dependency of the findings in this study would be helpful.**

Thank you for your comment, and for highlighting the new study by Cornford et al. (2020). Based on your comments and those provided by reviewers 1 and 2, we have expanded Sect. 2 to better justify the validity of the SSA approximation, clarify the boundary conditions used in our model, as well as its limitations compared to a Full Stokes model. We also note that Úa has been previously used to model paleo ice streams (Jones et al., 2021). The start of Sect. 2 now reads:

[revised manuscript text omitted]

Ito, K., & Kunisch, K. (2008). *Lagrange multiplier approach to variational problems and applications*. Society for Industrial and Applied Mathematics.

Wirbel, A., & Jarosch, A. H. (2020). Inequality-constrained free-surface evolution in a full Stokes ice flow model *(evolve_glacier v1.1)*. *Geoscientific Model Development*, *13*, 6425-6445.

**2.** **Context of findings: The authors assert throughout the paper that "ice sheet models overestimate ice surface elevations and underestimate the pace of ice sheet melt contributing to sea level rise compared to empirical reconstructions." I find this to be an oversimplification for a number of reasons. There are many examples of ice sheet models that fit empirical constraints during glacial and interglacial periods (e.g. Whitehouse et al., 2012; Golledge et al., 2013; Goezler et al., 2016; Whitehouse et al., 2017; Clark et al., 2020), as well as the rate of mass loss during deglaciations and through the Holocene (e.g. Gomez et al., 2013; Lecavalier et al., 2014; Tigchelaar et al., 2018; Cuzzone et al., 2019; Briner et al., 2020; Albrecht et al., 2020a). While two studies are cited with records from the Transantarctic Mountains (Lines 72-73), even models analysed within those studies show similar rates of ice thinning as observed in the records. I agree with the authors that datamodel mismatches occur in both Greenland and Antarctica, particularly with respect to the timing of regional ice thinning, though I'm not sure that the findings here definitively rule out other possible explanations, such as uncertainty in climate forcing (e.g. Lowry et al., 2019; Albrecht et al., 2020a), or uncertainty in the surface exposure ages themselves (e.g. Jones et al., 2019). It would be more accurate to simply point out that many previous paleo-ice sheet model studies, which use coarse resolution, do not resolve**

**site-specific features and this could contribute to observed timing mismatches between surface exposure records and model simulations.**

We agree with the reviewer, and acknowledge that the critique of previous ice sheet modelling should be clearer. We have rephrased the last paragraph of the introduction to reflect the points made by the reviewer, and clarify that by 'empirical constraints' we were referring to cosmogenic nuclide dating:

*"While ice sheet models of Greenland and Antarctica have been able to broadly fit ice geometries reconstructed from empirical data, including the approximate rates of ice thinning that are recorded by cosmogenic exposure ages (Whitehouse et al., 2012; Briggs et al., 2014; Albrecht et al., 2020), most models struggle to replicate the inferred timing of ice thickness change (Jones et al., 2020; Stutz et al., 2020; Johnson et al., 2021). Such data-model mismatches are likely due to a combination of factors, one of which is the spatial resolution of the models (Lowry et al, 2020; Johnson et al., 2021). When run over a glacial-interglacial cycle, ice sheet models do not typically resolve the pattern of ice flow around individual nunataks, and consequently cannot resolve the transient response of the ice surface at the sampled locations."*

We also simplified our discussion in line with the reviewer's suggestion:

*"Experiments using a regular grid at these resolutions do not resolve site-specific ice surface elevation responses, which contributes to mismatches relative to reconstructions from cosmogenic exposure dating (Spector et al., 2019; Stutz et al., 2020; Johnson et al., 2021)."*

**References not previously mentioned:**

Briggs, R. D., Pollard, D., & Tarasov, L. (2014). A data-constrained large ensemble analysis of Antarctic evolution since the Eemian. *Quaternary Science Reviews*, *103*, 91-115. doi: https://doi.org/10.1016/j.quascirev.2014.09.003

Lowry, D. P., Golledge, N. R., Bertler, N. A., Jones, R. S., McKay, R., & Stutz, J. (2020). Geologic controls on ice sheet sensitivity to deglacial climate forcing in the Ross Embayment, Antarctica. *Quaternary Science Advances*, *1*, 100002. doi: https://doi.org/10.1016/j.qsa.2020.100002

Johnson, J. S., Pollard, D., Whitehouse, P. L., Roberts, S. J., Rood, D. H., & Schaefer, J. M. (2021). Comparing glacial-geological evidence and model simulations of ice sheet change since the last glacial period in the Amundsen Sea sector of Antarctica. *Journal of Geophysical Research: Earth Surface*, e2020JF005827.doi: https://doi.org/10.1029/2020JF005827

**3.** **Influence of GIA: I am surprised that glacial isostatic adjustment was not considered in the simulations given the timescale. The calculation of surface exposure ages is highly dependent on the elevation, and correcting for GIA-driven elevation changes in non-trivial (Jones et al., 2019). Ice sheet models themselves are highly sensitive to GIA because it impacts bed elevation (e.g., Gomez et al., 2013; Kingslake et al., 2018; Colleoni et al., 2018; Albrecht et al., 2020b). While this study uses idealized experiments, in reality the basal topography would evolve through time in response to changes in ice loading. The authors do provide a useful discussion on limitations in the simplified SMB forcing for the idealized experiments (Lines 320-334), but a similar discussion with respect to GIA is currently lacking.**

We have now included a discussion about the role of GIA, together with our discussion on the simplified SMB forcing. We  highlight that in Antarctica, GIA is more likely to play a regional role, rather than add substantial variations between the upstream and downstream faces of a nunatak:

*"We do not consider glacial isostatic adjustment (GIA) in our experiments. Bedrock topography evolves through time due to changes in ice loading, which can influence the sensitivity of the ice sheet to sea level forcing, potentially impacting the grounding line position (Whitehouse et al., 2019), and the cosmogenic production rate, possibly impacting the calculated exposure ages (Jones et al., 2019). In terms of patterns of exposure up and downstream of nunataks, the influence of differential isostatic bedrock uplift rates on each side of the nunatak are much smaller than the changes in ice surface elevation considered here, since bedrock elevation changes due to GIA happen at a spatial scale larger than the ~60 km observed in our experiments (e.g., Ivins and James, 2005)"*

**References not previously in the manuscript:**

Ivins, E., & James, T. (2005). Antarctic glacial isostatic adjustment: A new assessment. *Antarctic Science, 17*(4), 541-553. doi:10.1017/S0954102005002968

Jones, R. S., Whitehouse, P. L., Bentley, M. J., Small, D., & Dalton, A. S. (2019). Impact of glacial isostatic adjustment on cosmogenic surface-exposure dating. Quaternary Science Reviews, 212, 206-212. doi: https://doi.org/10.1016/j.quascirev.2019.03.012

Whitehouse, P. L., Gomez, N., King, M. A., & Wiens, D. A. (2019). Solid Earth change and the evolution of the Antarctic Ice Sheet. *Nature communications*, *10*(1), 1-14. doi: https://doi.org/10.1038/s41467-018-08068-y

**Specific comments:**

**4.** **Line 16-17: See above general comment on context of findings.**

We have rephrased this passage:

*"Such differences, however, are not typically captured by continent-wide ice sheet models due to their coarse grid resolutions. Their inability to capture site-specific surface elevation changes appears to be a key reason for the observed mismatches between the timing of ice free conditions from cosmogenic exposure dating and model simulations."*

**5.** **Fig 2: Is there a scale for the slices? The colours correspond to upstream and downstream, correct? The point is to show that samples are taken in nearly all directions?**

The reviewer is correct. Following the suggestion by Reviewer 1 (comment 12), we have removed the colour coding, as it added no new information and was distracting.

**6.** **Line 136: citation? Does SMB only change in one direction?**

We have added a reference to Agosta et al. (2019, The Cryosphere), who show that most regions where nunataks are present are not suffering ablation at present. As for SMB changes, they only happen in one direction (i.e., increased ablation). This is shown in Fig. 3a,c, and we made it clearer at the beginning of Sect. 2.3:

*"In order to understand whether ice thinning occurs uniformly up and downstream of a nunatak, we impose three different degrees of ice thinning uniformly over the domain. The perturbations to SMB that induce ice thinning are applied through b(t) in Eq. (2), and only evolve in one direction over the entire domain, towards increased ablation."*

**7.** **Line 160: Any sensitivity tests for this value? 5-6% K-1 is appropriate for EAIS (Frieler et al. 2015). WAIS is less straightforward (Fudge et al. 2018).**

We believe our text lacked clarity, which might have confused the reviewer. The percentages shown refer to the weight of the SMB factor applied in b(t). Using the 'thw' scenario as an example, the value for b(0) is $0.08*-0.75$ m $a^{-1}$, while b(20 kyr) is $1.0*-0.75$ m $a^{-1}$. As originally stated in L161-163, these values were chosen so that the total ice thinning matches the thinning at the respective regions in a deglacial model experiment by Golledge et al. (2014). We have added the example above to the text in Sect. 2.1 to make it clearer.

**8.** **Fig 4: Could you make the inset larger? It is difficult to see the coloured circles that correspond to panels c and d.**

Following the changes based on the suggestion by Reviewer 1 (comment 15), we have removed the inset, and instead added this information to the new figure, presented below, which will be incorporated to the manuscript:

[Figure]

**New Figure.** Bedrock elevation (m a.s.l.) used in the experiments where the nunatak is placed (a) transverse and (b) parallel to ice flow. Ice flow in this figure is from left to right.  Transect lines show the position of the transects presented in Fig. 4a,b, and coloured circles show the locations for the ice surface evolution analysis presented in Fig. 4c,d following the same colours as the lines therein.

We thank the reviewer for the thorough and insightful review of our manuscript, and look forward to the opportunity to present a significantly updated manuscript based on the revisions outlined above.

---

## Author Response (AR2)

Dear Alex,

Thanks for revising our manuscript and providing additional feedback. Below we show how the final comments were addressed.

Best regards,

Martim

**Your revised manuscript is in very good shape. I find it is near ready for publication, with only the minor changes below needed:**

**L23, and elsewhere: sea level rise => sea-level rise**
Changed as suggested.

**L47: the complex subglacial terrain => complex subglacial terrain**
Changed as suggested.

**L51: from where => from which**
Changed as suggested.

**L60: up and downstream => up- and downstream**
Changed as suggested.

**L106: sliding law issues => sliding-law issues**
Changed as suggested.

**L134: Unclear phrasing, please revise: "As nunataks near the Antarctic coast or along the margins of palaeo ice streams are often situated ..."**
Rephrased as: "Because nunataks close to the Antarctic coast, or along the margins of palaeo ice streams, are often situated in proximity to over-deepened troughs..."

**L155, Eq. 2: Consider modifying equation such that the variable "SMB" is represented by one letter, which is clearer and commonly used. Since you already use "b(t)", why not b(x,t) = b_0 - (b_0-b_e)/L_x*x + b_anom(t), for**

**example?**

We have followed the suggestion, but using "a" as opposed to "b", thus keeping the time-dependent term as "b(t)". We also made the necessary changes in the text (L154, L156 – 158, and Table 1)

**L161: Model spin up => The model spin up**

Changed as suggested.

**L316: that this a => that this is a**

The typo was fixed.

**L340: but simplify => but they also simplify**

Changed as suggested.